# PLGA-particle vaccine carrying TLR3/RIG-I ligand Riboxxim synergizes with immune checkpoint blockade for effective anti-cancer immunotherapy

Julia Koerner [1,9], Dennis Horvath [1,2,9], Valerie L. Herrmann[1,8], Anna MacKerracher[1], Bruno Gander[3], Hideo Yagita[4], Jacques Rohayem[5,6] & Marcus Groettrup [1,2,7 ✉]

With emerging supremacy, cancer immunotherapy has evolved as a promising therapeutic modality compared to conventional antitumor therapies. Cancer immunotherapy composed of biodegradable poly(lactic-co-glycolic acid) (PLGA) particles containing antigens and toll-like receptor ligands induces vigorous antitumor immune responses in vivo. Here, we demonstrate the supreme adjuvant effect of the recently developed and pharmaceutically defined double-stranded (ds)RNA adjuvant Riboxxim especially when incorporated into PLGA particles. Encapsulation of Riboxxim together with antigens potently activates murine and human dendritic cells, and elevated tumor-specific CD8[+] T cell responses are superior to those obtained using classical dsRNA analogues. This PLGA particle vaccine affords primary tumor growth retardation, prevention of metastases, and prolonged survival in preclinical tumor models. Its advantageous therapeutic potency was further enhanced by immune checkpoint blockade that resulted in reinvigoration of cytotoxic T lymphocyte responses and tumor ablation. Thus, combining immune checkpoint blockade with immunotherapy based on Riboxxim-bearing PLGA particles strongly increases its efficacy.

[1] Division of Immunology, Department of Biology, University of Konstanz, Konstanz, Germany. [2] Centre for the Advanced Study of Collective Behaviour, University of Konstanz, Konstanz, Germany. [3] Institute of Pharmaceutical Sciences, ETH Zürich, Zürich, Switzerland. [4] Department of Immunology, Juntendo University School of Medicine, Tokyo, Japan. [5] Riboxx GmbH, BioInnovationszentrum, Dresden, Germany. [6] Institute of Virology, Medical Faculty Carl Gustav Carus, Dresden University of Technology, Dresden, Germany. [7] Biotechnology Institute Thurgau at the University of Konstanz (BITg), Kreuzlingen, Switzerland. [8]Present address: Boehringer Ingelheim Pharma, Cancer Immunology + Immune Modulation, Biberach/ Riß, Germany. [9]These authors contributed equally: Julia Koerner, Dennis Horvath. ✉email: Marcus.Groettrup@uni-konstanz.de

C ancer still represents one of the most prevalent and challenging malignancies worldwide as a failure of standard therapies results in tumor relapse and metastasis formation. Hence, there is an urgent need for the development of alternative therapies such as immunotherapy with cancer vaccines[1]. Due to the unique ability of dendritic cells (DCs) to prime and activate naive T cells, effective antigen charging and stimulation of DCs are key goals in immunotherapy. We and others have established the use of PLGA-MP (microparticles) and NP (nanoparticles) as efficient antigen delivery systems for targeting DCs in situ[2]. The aliphatic copolymer PLGA is extensively used for optimizing controlled and joint release of immune stimulants and antigens in cancer immunotherapy in preclinical settings[3]. Due to its biodegradability and proven safety profile, PLGA and its formulations have been licensed by the U.S. Food and Drug Administration (FDA) and European Medicines Agency (EMA) for pharmaceutical applications via parenteral and mucosal routes[4]. Especially, PLGA-MP exhibit ideal properties for facilitated uptake by antigen-presenting cells (APCs). Concomitant delivery of antigens and adjuvants to the same APC leads to efficient DC or macrophage activation and simultaneous activation of both, CD4+ T helper cells and CD8+ cytotoxic T lymphocytes (CTL), via cross-presentation[5].

Successful immunotherapy against cancer also requires an ideal adjuvant. Multiple immunostimulatory RNA duplexes with agonistic activity for pattern-recognition receptors (PRR) have been reported as candidates for adjuvants in cancer immunotherapy. Double-stranded RNA molecules classically trigger endosomal TLR3 (Toll-like receptor 3). Depending on the structural characteristics and length of the viral RNA mimetic, it may also activate cytosolic RIG-I-like receptors (RLRs), including retinoic acid-inducible gene I (RIG-I) or melanoma differentiation-associated gene 5 (MDA5). TLR3 and RIG-I downstream signaling results in nuclear translocation of the key transcription factors nuclear factor κ/light-chain enhancer of activated B cells (NF-κB) and interferon regulatory factor (IRF)3/7 culminating in secretion pro-inflammatory cytokines and of type I interferons (IFN), respectively. Co-encapsulation of TLR3 ligands with antigen greatly enhanced PLGA particle-mediated cancer vaccine efficacy by inducing reliable maturation of DCs[6]. The dsRNA analog poly(I:C) (polyinosinic:polycytidylic acid) is the most widely used TLR3 agonist but also engages RIG-I and preferably MDA5[7]. Its adjuvant properties result from induction of type I interferons and pro-inflammatory cytokines such as interleukin (IL)-6, tumor necrosis factor (TNF), and interleukin (IL)−1β[8] thus enabling an improved efficacy of cancer immunotherapy in murine tumor models[9–11]. However, its ill-defined macromolecular structure and heterogeneous composition prevented polyI:C from extensive clinical use due to unpredictable pharmacokinetics and toxicity issues. The TLR3/RIG-I ligand Riboxxim™, in contrast, is characterized by its well-defined chemical structure as double-stranded RNA and nucleotide composition of 100 bp for effective TLR3 triggering. An uncapped triphosphate moiety at the 5′ end of Riboxxim also enables the activation of RIG-I[12]. Riboxxim exhibits very good solubility in water, prolonged stability in solution at 4 °C and in serum.

Cancer immunotherapy has been greatly improved and promoted by pharmacological blockage of so-called immune checkpoint receptors such as CTLA-4 (cytotoxic T-lymphocyte-associated protein 4), PD-1 (programmed cell death protein 1), and its ligand PD-L1. These molecules can be co-opted as immune evasion mechanism by tumors to inhibit CTL activity and suppressing antitumor immune responses[13]. Monoclonal antibodies directed against these checkpoint receptors have achieved efficient remission of several tumor types, which resulted in FDA and EMA approval of immune checkpoint blockade

for adjuvant treatment of advanced melanoma, head and neck squamous cell carcinoma, and metastatic bladder carcinoma[14]. However, the majority of cancer patients did not benefit from this therapy with only up to 20% complete responses in patients with solid cancer. The therapeutic success of immune checkpoint blockade necessitates the preexistence of cancer-specific CTLs at the tumor site. Thus, there is an urgent medical need to develop defined immunotherapeutic approaches that would elicit tumor-specific CTLs and boost the success of checkpoint blockade.

Previously, we have shown that PLGA-MP mediated co-delivery of tumor antigens or lysates along with several experimental TLR ligands yielded potent and long-lasting anti-cancer immune responses in vivo affording suppression of tumor growth[15–18]. With respect to a possible clinical translation, this study demonstrates the immune-potentiating capacity of GMP-grade Riboxxim contained in PLGA-MP and the synergistic therapeutic effect of combining this approach with immune checkpoint blockade.

## Results

**Comparison of PLGA particles of different size.** While PLGA-MP have been produced by spray-drying, NPs were prepared for comparison by the standard double emulsion solvent evaporation method. Scanning electron microscopy (SEM) images show uniform spherical PLGA-NPs with a smooth surface of an average size of 250 nm in diameter. MPs produced by spray-drying possess a relatively broad size distribution ranging from nano-sized up to 1.5 μm sized particles, notoriously suitable for cellular uptake and internalization by different APCs (Fig. 1a, b). Since the site of vaccination greatly impacts the effectiveness of cancer vaccines, we analyzed the efficacy of tumor-specific CD8+ CTL induction with PLGA-NP or MP using different administration routes. PLGA particles containing ovalbumin (OVA) and GMP-certified pyrogen-free Riboxxim were injected either intraperitoneally (i.p.), intramuscularly (i.m.), intranodally (i.nd.), or subcutaneously (s.c.) into C57BL/6J mice. The maximal cytokine response on day 6 after vaccination was achieved after s.c. application (Fig. 1c, d). We decided to use PLGA-MP for further experiments, since overall CD8+ response rates to the OVA peptide SIINFEKL were slightly better for each application route (see also Supplementary Fig. 1c–e).

**Characteristics of Riboxxim-containing PLGA particles.** Physicochemical characterization of the different PLGA-MP formulations by dynamic light scattering (DLS) analysis showed unimodal size distribution with a mean particle size of ~1 μm in diameter and negative surface charge (Supplementary Fig. 2a and Table 1). Encapsulated amounts and release kinetics of OVA protein were similar between MP-OVA/polyI:C (MP-OVA/pIC) and MP-OVA/Riboxxim (MP-OVA/Rib) showing a typical burst release within 24 h followed by sustained release of protein over time (Fig. 2a, b). This initial burst is commonly attributed to adsorbed antigenic material on the particle surface that is desorbed upon particle dispersion in aqueous media. Although not differing in encapsulation efficiency, the OVA release profile of PLGA-MP was superior compared to nanoparticles (Supplementary Fig. 1a, b). The used PLGA type RG502H almost completely degrades within 30 days as visualized by fluorescent MP-OVA/Rib in vivo (Fig. 2c). The antigen accumulation at the site of injection fostered constant PLGA-MP trafficking to popliteal and inguinal lymph nodes by skin resident DCs (Fig. 2d). This depot effect prolongs T-cell stimulation. Both, DCs as well as macrophages engulfed PLGA-MP-OVA/Rib efficiently and to a similar extent as illustrated by approximately 60% of QD (Quantum-Dot)-positive staining of primary murine and human cells and

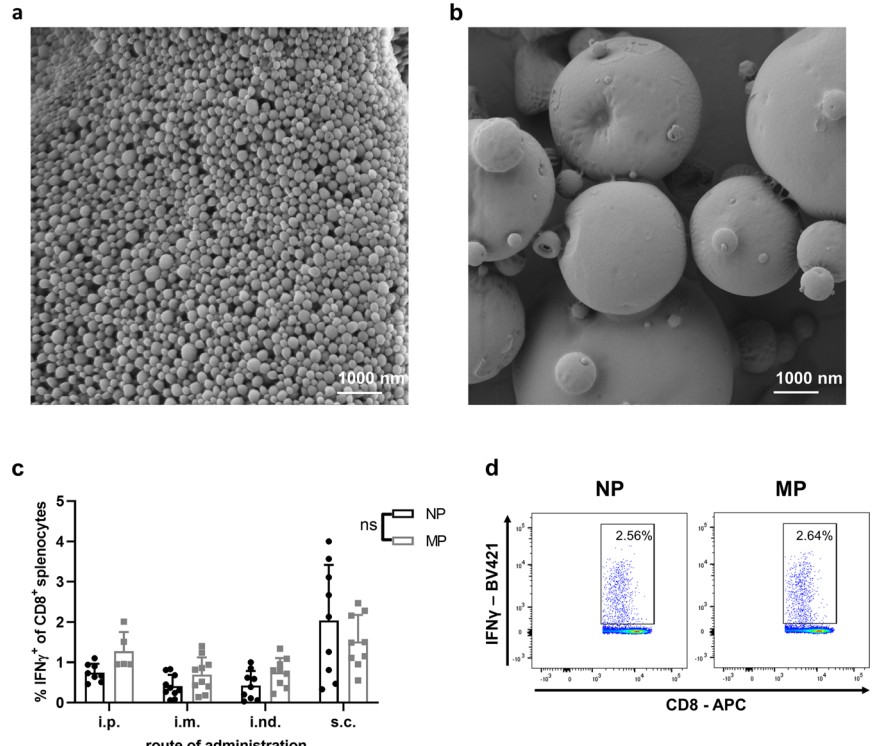

**Fig. 1 Nano- or micron-sized PLGA particles containing ovalbumin (OVA) and Riboxxim are potent vaccine delivery systems for vaccination. a, b** Size distribution and morphology of nanoparticles (**a**) and microparticles (**b**) containing encapsulated OVA and Riboxxim were analyzed by scanning electron microscopy (SEM). Scale bars, 1000 nm. SEM images were acquired from three different particle batches with similar results. **c** C57BL/6J mice were immunized with nanoparticles (NP, black dots) or microparticles (MP, gray squares) charged with OVA protein (250 µg/mouse) and Riboxxim (2.5 µg/mouse) via the intraperitoneal (i.p., $n = 8$ for NP, $n = 5$ for MP), intramuscular (i.m., $n = 10$ for NP, $n = 10$ for MP), intranodal (i.nd., $n = 9$ for NP, $n = 9$ for MP), or subcutaneous (s.c., $n = 9$ for NP, $n = 9$ for MP) route demonstrating the latter as most efficient. Six days post-immunization, an intracellular cytokine staining for IFNγ$^+$ of CD8$^+$ splenocytes was performed and analyzed via flow cytometry. Statistics: two-way ANOVA followed by Šídák's multiple comparisons test with ns, not significant. Data are presented as means ± SD and represent pooled data from three independent experiments. **d** A representative dot plot showing the frequency of IFNγ$^+$CD8$^+$ splenocytes for indicated treatment groups.

**Table 1 Particle size, polydispersity index (PDI), and ζ-potential of PLGA micro- and nanoparticles.**

| Vaccine compound | Size (nm ± SD) | PDI ± SD | ζ potential (mV ± SD) |
|---|---|---|---|
| MP-OVA/polyI:C | 1377 ± 62.07 | 0.256 ± 0.102 | −47.3 ± 1.02 |
| MP-OVA/Riboxxim | 1437 ± 162.9 | 0.396 ± 0.129 | −51.0 ± 2.85 |
| MP Empty | 1436 ± 235.2 | 0.385 ± 0.172 | −56.0 ± 0.608 |
| NP OVA/Riboxxim | 240.4 ± 24.56 | 0.407 ± 0.128 | −30.2 ± 0.692 |

Results are expressed as means ± SD. For each compound, at least three independent batches were prepared. Measurement was performed in triplicates ($n = 3$).
*MP* microparticles, *NP* nanoparticles, *SD* standard deviation, *PDI* polydispersity index.

various cell lines after 6 hours of MP pulsing (Fig. 2e and Supplementary Fig. 2b). Particle uptake was verified by mouse CD11c$^+$ bone marrow-derived DCs (BMDCs) and peritoneal macrophages (pMØ), as well as by human CD14$^+$ monocytes, that have been in vitro differentiated into monocyte-derived DCs (MoDCs) or macrophages (MoMØ). Efficient particle engulfment of about 35% could also be demonstrated with CD1c$^+$ and CD141$^+$ primary human myeloid dendritic cells isolated from peripheral blood (mDCs). Of note, no in vitro cytotoxicity was detected after engulfment of PLGA-MP by BMDCs over several days (Supplementary Fig. 2c).

**Riboxxim induces increased DC maturation and type I IFN cytokine secretion.** A proper activation of DCs is crucial for T-cell targeted cancer vaccines. Whether Riboxxim promotes upregulation of maturation markers and the release of cytokines by TLR3-expressing murine and human DCs was analyzed in comparison to conventional polyI:C (pIC) or a pyrogen-free polyI:C variant (pIC pyf). Endotoxin levels of ≤0.5 EU/ml further demonstrated safety of Riboxxim, while the most widely used conventional polyI:C exhibited a high level of endotoxin and pyrogen contaminations, which leads to a potential contribution of TLR4 signaling after addition of polyI:C (Supplementary Fig. 3b). Stimulation of murine DCs with Riboxxim enhanced upregulation of co-stimulatory molecules and major histocompatibility complex (MHC) class I, which was even superior to the upregulation by polyI:C. (Fig. 3a). As stimulation of human CD14$^+$ MoDCs did not result in any DC activation, we searched for an alternative human DC model. Differentiation of the human monocytic cell line THP-1 into MoDC-like DCs resulted in potent DC activation by Riboxxim, which was similar if not superior to the activation by polyI:C (Supplementary Fig. 3c). Compared to a prominent expression of TLR3 in murine CD8α$^+$ cDC1[19], in vitro differentiated human MoDCs possess only hardly detectable levels of TLR3 and donor-dependent variations in RIG-I and MDA5 expression levels[20]. The highest TLR3 expression in human primary DC subsets occurs in myeloid BDCA1$^+$ (blood DC antigen 1, CD1c$^+$) and BDCA3$^+$ (CD141$^+$) cells, found in low numbers in human peripheral blood[21].

When analyzing these cells isolated from human peripheral blood mononuclear cells (PBMCs), stimulation with Riboxxim achieved efficient elevation of co-stimulatory molecules, maturation markers, and MHC-I equivalent to the stimulation

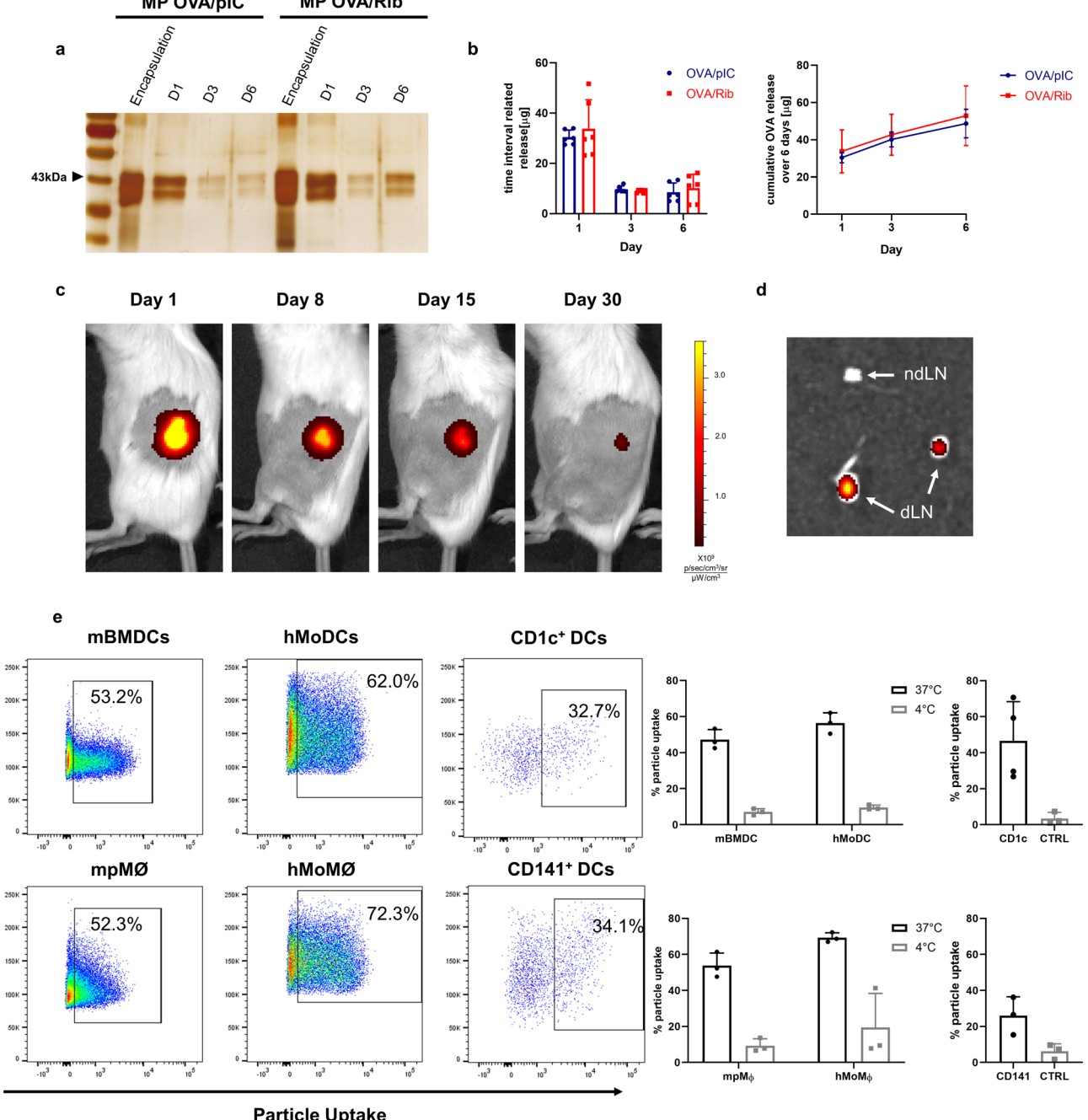

**Fig. 2 PLGA-MP exhibit ideal release profiles and are efficiently engulfed by human and mouse dendritic cells. a** 5 mg MP containing 250 μg OVA and 2.5 μg polyI:C (MP-OVA/pIC) or 2.5 μg Riboxxim (MP-OVA/Rib) were incubated in PBS (pH 7.4) at 37 °C for a total period of 6 days. Amounts of encapsulated OVA and of OVA released from PLGA-MP at indicated time points were determined by SDS-PAGE and subsequent silver staining. Data are presented as means ± SD from three independent experiments. **b** Encapsulation efficiency and cumulative in vitro OVA release from 5 mg PLGA-MP-OVA/polyI:C (OVA/pIC, blue circles, $n = 6$) or PLGA-MP-OVA/Riboxxim (OVA/Rib, red squares, $n = 6$) over 6 days as analyzed by MicroBCA™ assay. **c** Near-infrared (NIR) fluorescence IVIS® images of a representative BALB/c mouse s.c. injected with 5 mg fluorescent MP-QD705/OVA/Rib at the indicated days post-immunization. Scaling of the pseudo-color code is depicted next to the luminescent images. **d** Ex vivo NIR fluorescence IVIS® image of draining popliteal and inguinal LN (dLNs) and of the contralateral non-draining LN (ndLN) of a representative mouse 6 days after s.c. immunization with MP-QD705/OVA/Rib. **e** Percentage of fluorescent particle internalization 6 h after incubating DCs with 10 μg/ml MP-QD705/OVA/Rib at 37 °C (black circles) or 4 °C (no uptake control, gray squares) was assessed by flow cytometry. Approximately 60% of PLGA-MP were efficiently engulfed by mouse bone marrow-derived DCs (mBMDCs, $n = 3$) or thioglycolate elicited peritoneal macrophages (mpMØ, $n = 3$), as well as by in vitro differentiated human monocyte-derived DCs (hMoDCs, $n = 3$) or monocyte-derived macrophages (hMoMØ, $n = 3$). PLGA particle uptake by human myeloid DCs was assessed using flow cytometry by gating on QD705+CD1c+ ($n = 4$) and QD705+CD141+ cells ($n = 3$) (black circles) after incubation of fluorescent microparticles with $10^7$ CD14-negative PBMCs. CD1c-negative ($n = 4$) or CD141-negative cells ($n = 3$) served as control (CTRL, gray squares). Data are presented as means ± SD and are derived from at least two independent experiments with a similar outcome.

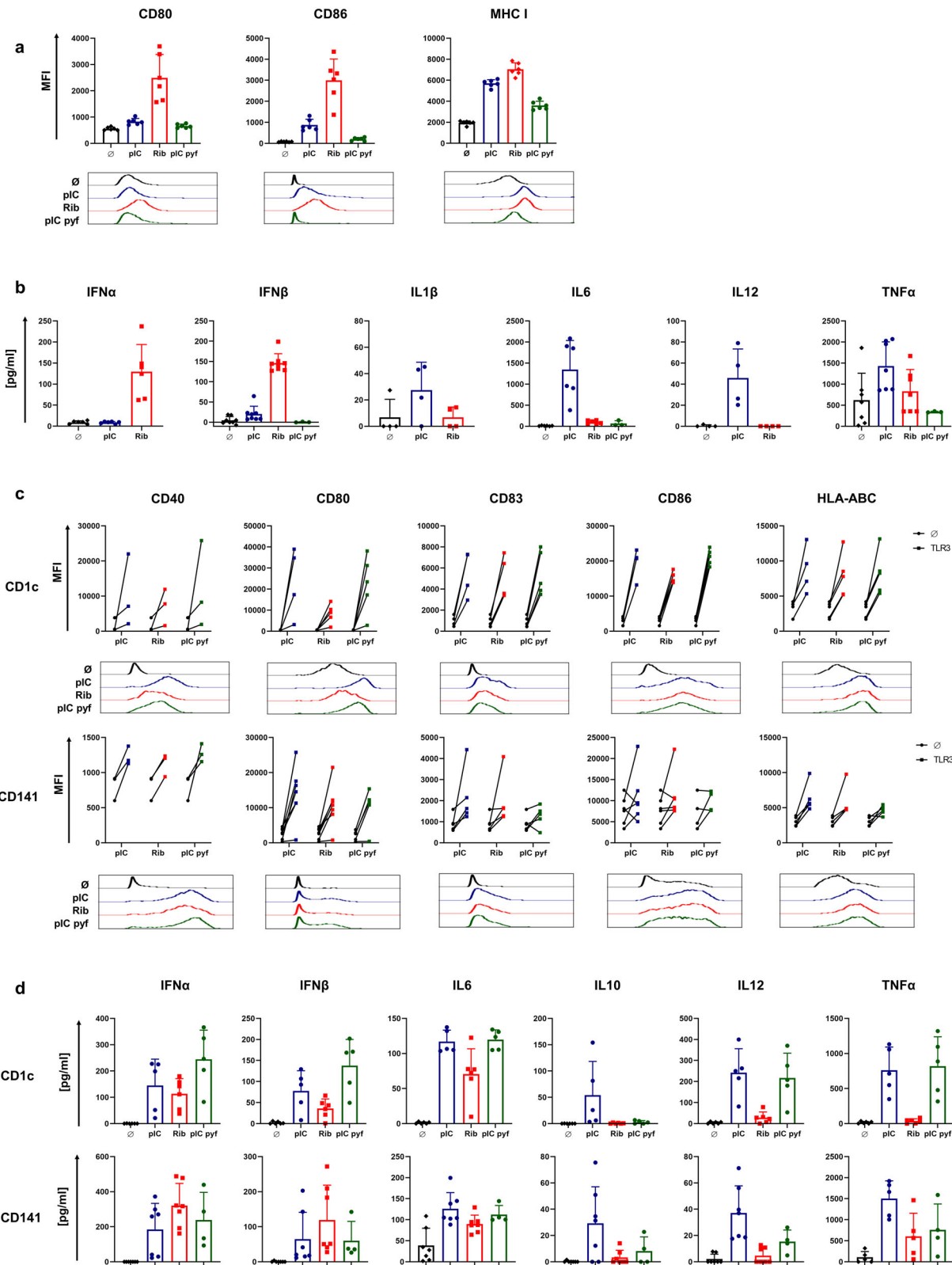

by polyI:C (Fig. 3c). This result is pivotal since BDCA3[+] DCs are the most relevant target of cancer vaccines due to their superior capacity to cross-present tumor antigens to CD8[+] T cells[21]. The advantage of encapsulating Riboxxim into PLGA relies on its intracellular trafficking into the endosome, where TLR3 is located. Consistently, MP-Riboxxim efficiently induced expression of DC activation and maturation marker in human

monocytic-like DCs and myeloid DCs (Supplementary Fig. 3c, d). Interestingly, the upregulation of co-stimulatory molecules was even increased after incubation of mouse DCs or in vitro-induced MoDCs with MP-Riboxxim in presence of protein or peptide antigen (Supplementary Fig. 3a, c). Notably, mouse DC activation via MP-OVA/pI:C was negligible (Supplementary Fig. 3a).

**Fig. 3 Riboxxim induces potent dendritic cell activation and secretion of type I interferons. a**, **c** Mouse BMDCs (**a**, **b**) or freshly isolated CD1c[+] and CD141[+] mDCs from human PBMCs of 15 healthy donors (**c**, **d**) were incubated in vitro with 20 µg/ml polyI:C (pI:C, blue circles), Riboxxim (Rib, red squares) or pyrogen-free polyI:C (pI:C pyf, green circles) for 48 h or left untreated (∅, black diamonds). DC maturation and activation levels were assessed by flow cytometric analysis of the indicated surface markers. Mouse data (**a**) are presented as the mean fluorescence intensity (MFI) ± SD ($n = 6$). Human data (**c**) are presented as changes in MFI of treated (squares, TLR3) to unstimulated control cells (black circles, ∅) for individual donors ($n = 3–5$ for CD1c, $n = 3–7$ for CD141). Representative flow cytometry histograms for every marker and treatment are shown below the graphs in respective colors. **b**, **d** ELISA analysis of type I IFN and pro-inflammatory cytokine production in culture supernatants of mouse BMDCs ($n = 3–7$) (**c**) and human mDCs ($n = 5–6$ for CD1c[+], $n = 4–7$ for CD141[+]) (**d**) with the indicated treatments. All data are presented as means ± SD. Graphs represent pooled data from at least three independent experiments with a similar outcome.

Riboxxim stimulates the production of pro-inflammatory IL-6 by CD1c[+] and CD141[+] DCs (Fig. 3d). Interestingly, compared to poly(I:C), Riboxxim has a greatly enhanced capacity to promote type I IFN production in mouse but also in human primary DCs (Fig. 3b, d), which is of central importance for direct antitumor effects and proliferation and differentiation of antigen-specific CTLs[22]. Especially in mouse DCs, secretion of IFNα and IFNβ was not detected after polyI:C stimulation, suggesting a Riboxxim-specific cytokine profile. Interestingly, conventional polyI:C induced high levels of non-specific, pro-inflammatory cytokines IL-1β, TNF, IL-6, as well as immunosuppressive IL-10, which was also detectable using a pyrogen-free polyI:C variant—especially in human CD1c[+] cells (Fig. 3b, d). To address the significance of the TLR3/TICAM-1 and RIG-/MAVS pathway for the induction of DC activation by Riboxxim, BMDCs from wild-type (WT), $Tlr3^{-/-}$ or $Mavs^{-/-}$ mice were stimulated with external addition of Riboxxim or polyI:C (Supplementary Fig. 4). $Mavs$-deficient DCs showed a slightly impaired upregulation of both co-stimulatory molecules and maturation markers after administration of Riboxxim (Supplementary Fig. 4b). TLR3-dependent upregulation of these markers was comparable to Riboxxim stimulation in WT cells. BMDCs deficient for the adapter molecule MAVS failed to release type I IFN with Riboxxim (Supplementary Fig. 4a), suggesting the importance of RIG-I/MAVS activity for production of IFNα/β, which is enabled by the 5′-triphosphate group of Riboxxim. These data demonstrate that both, endosomal TLR3 activation and MAVS-mediated signaling is indispensable for potent DC activation and maturation, as well as specific type I IFN cytokine production in response to stimulation with Riboxxim. Importantly, immunization with PLGA-MPs did not elicit a serum cytokine response post-vaccination (Supplementary Fig. 5b) or histological changes or damages in major organs (Supplementary Fig. 5a). Thus, PLGA-MP administration is safe and biocompatible for controlled drug delivery in antitumor vaccination.

Collectively, Riboxxim exhibits enhanced efficacy of upregulation of MHC class I and co-stimulatory molecules on the mouse and human DCs supported by a distinct release of type I IFNs, which enforces improved antigen-presentation and priming of CTLs[23].

**Improved priming and activation of CD8[+] T cells by PLGA-MP encapsulated Riboxxim.** In addition to enhanced antigen presentation, co-encapsulating Riboxxim also translated into improved antigen-specific CD8[+] T-cell priming as visualized by increased intracellular IFNγ staining and IFNγ secretion (Fig. 4a, b and Supplementary Fig. 6a, b). C57BL/6J mice were vaccinated with titrated amounts of dsRNA adjuvants in OVA-containing microparticles (Fig. 4a) or different amounts of PLGA-MP-OVA/TLR3 particles per mouse (Fig. 4b) and sacrificed 6 days later to determine activation of OVA-specific effector T cells in the spleen. MP-OVA/Riboxxim elicited significantly more IFNγ-producing antigen-specific T cells than MP-OVA/polyI:C—

irrespective of the adjuvant dose or the employed amount of PLGA vaccine.

We also analyzed the cytotoxic potential of generated CD8[+] T cells. Mice that were immunized with MP-OVA/Riboxxim showed a more pronounced peptide-specific killing as compared to MP-OVA/polyI:C, and immunization was effective even at low doses of Riboxxim, as illustrated by complete OVA-specific cytolysis of target cells in vivo (Fig. 4c). Relating to the depot effect and sustained antigen release after subcutaneous administration of PLGA-MP, we investigated the duration of the PLGA-MP-mediated immune response (Fig. 4d and Supplementary Fig. 6c). A single vaccination of C57BL/6J mice with PLGA-MP-OVA/Riboxxim led to potent CD8[+] T-cell responses that lasted at least 6 weeks and were significantly stronger than those elicited by PLGA-MP-OVA/polyI:C. Moreover, MP containing OVA/Riboxxim outperformed MP-OVA/polyI:C in routing encapsulated antigens into class I and II processing and cross-presentation to CD8[+] T cells (Supplementary Fig. 6d). The proliferation of OVA-specific OT-1 T cells was greatly enhanced after stimulation with MP-OVA/Riboxxim, illustrated by proliferation assays in vitro (Fig. 4f and Supplementary Fig. 6e) and in vivo (Fig. 4e). Consistent with TLR3 and RIG-I-dependent DC activation described above, DC-mediated generation of improved antigen-specific CD8[+] T-cell responses by encapsulation of Riboxxim relies on combined activation of TLR3/TICAM-1 and MAVS signaling. IFNγ production of ex vivo re-stimulated splenocytes from MP-OVA/Riboxxim-vaccinated mice was similarly reduced in $Mavs^{-/-}$ and $Tlr3^{-/-}$ mice compared to the IFNγ response in WT mice. On the other hand, CTL responses to MP-OVA/polyI:C were primarily blunted in $Tlr3$-deficient animals (Supplementary Fig. 7). These data illustrate the need for activation of both signaling pathways to generate superior immune responses.

Additionally, we have investigated PLGA particle-elicited CTL responses to several human tumor peptide antigens of different cancers. Previously, we have shown that encapsulation of the HLA-A*0201 restricted immunodominant prostate cancer (PCa) epitope STEAP1$_{262–270}$ (six transmembrane epithelial antigen of the prostate) into PLGA microparticles enabled the induction of peptide-specific CTL activation and cytotoxic effector function in chimeric human HLA-A*0201 transgenic AAD mice[18]. Encapsulation of STEAP1$_{262–270}$ into PLGA-MP together with Riboxxim enabled even more potent CTL responses (Fig. 4g and Supplementary Fig. 6f). Notably, even more vigorous IFNγ responses were generated against further PCa epitopes, namely STEAP1$_{292–300}$, PSMA$_{469–477}$, and especially PAP$_{135–143}$ as well as the well-known cancer-testis antigen NY-ESO1$_{157–65}$ expressed in numerous cancer types. CTL response were additionally generated after immunization with PLGA particles carrying the breast cancer-peptide antigen HER2$_{689–297}$ or the melanoma antigen TRP-2$_{180–188}$. CD8[+] T-cell responses against these self-antigens were significantly increased in presence of Riboxxim, suggesting a superior ability to break immune tolerance when using Riboxxim as an immune adjuvant.

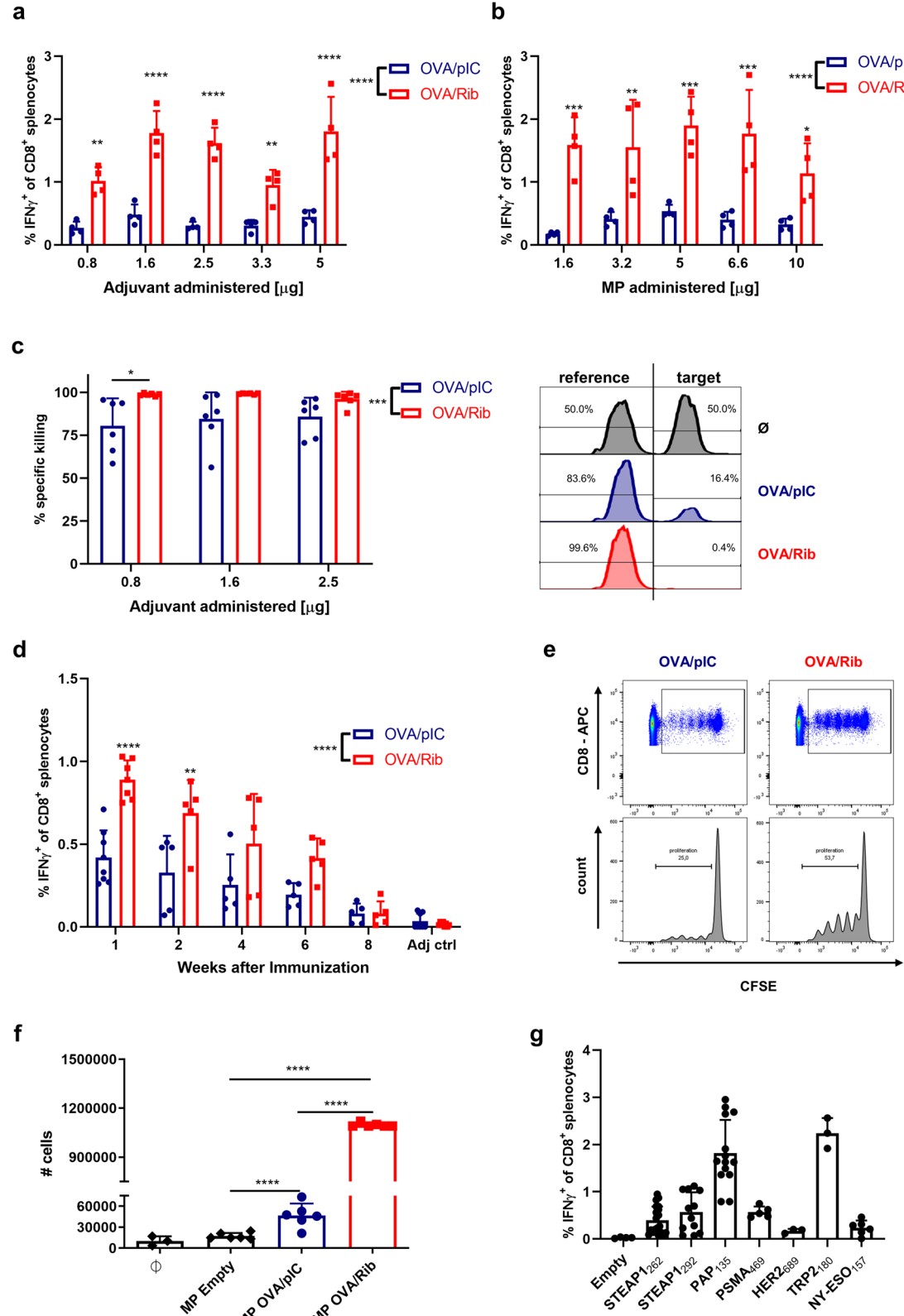

**Riboxxim induces advantageous antitumor responses that are synergistically enhanced by immune checkpoint blockade to enable tumor ablation**. To investigate the potential of antigen and Riboxxim-containing PLGA-MP for tumor immunotherapy we first evaluated the protective effect against OVA-expressing E.G7 thymomas, which were inoculated into the flanks of C57BL/6J mice (Supplementary Fig. 8a). Animals that had received a single vaccination of MP-OVA/Riboxxim or MP-OVA/polyI:C remained tumor-free throughout the whole experiment of 60 days (Fig. 5a, b). We then evaluated the therapeutic potential of PLGA-MP immunization of already established tumors. PLGA-MP treatment of the indicated groups was initiated when E.G7 tumor nodules were manually detectable (Fig. 5c, d). With the help of bioluminescent imaging, reduction of tumor mass was observed starting 8 days

**Fig. 4 PLGA-MP containing co-encapsulated tumor antigens and Riboxxim efficiently induce CD8[+] T-cell proliferation, increased tumor-specific T-cell responses, and peptide-specific in vivo cytotoxicity.** Analysis of immune responses against PLGA-microparticles (MP) containing OVA (**a–f**) or tumor-associated antigen (TAA)-epitopes (**g**) and polyI:C (pIC, blue circles) or Riboxxim (Rib, red squares). **a**, **b** For adjuvant and particle dose titration, C57BL/6J mice (n = 4) were immunized s.c. with 5 mg MP charged with 250 µg OVA and either different amounts of dsRNA analogs per MP (**a**) or varying amounts of MP-OVA-polyI:C (OVA/pIC, blue circles, n = 4) or MP-OVA/Riboxxim (OVA/Rib, red squares, n = 4) per mouse (**b**). Six days post-immunization, CTL activation was monitored by ICS for IFNγ[+] of CD8[+] splenocytes. Statistics: two-way ANOVA followed by Šídák's multiple comparisons test. *P < 0.05; **P < 0.01; ***P < 0.001; ****P < 0.0001. **c** Improved CTL-mediated cytotoxicity after vaccination of C57BL/6J mice with MP loaded with 250 µg OVA and different amounts of polyI:C (OVA/pIC, blue circles, n = 6) compared to different amounts of Riboxxim (OVA/Rib, red squares, n = 6) was monitored in vivo by specific lysis of CFSE-labeled, SIINFEKL peptide-loaded target cells. Mice immunized with MP-Empty (Ø, black, n = 3) served as a control for antigen-specific killing. Representative flow cytometry histograms of peptide-pulsed target cells and unpulsed reference cells are shown on the right. Statistics: two-way ANOVA followed by Šídák's multiple comparisons test. *P < 0.05; ***P < 0.001. **d** Kinetics of antigen-specific immune responses after vaccination of C57BL/6J mice (n = 5–8 for OVA/pIC, n = 5–7 for OVA/Rib) with 5 mg MP charged with 250 µg OVA and 2.5 µg dsRNA analogs was monitored by ICS for IFNγ[+] of CD8[+] splenocytes at different time points after immunization. PLGA-MP containing dsRNA adjuvants alone served as control (Adj ctrl, adjuvant control, n = 8). Statistics: two-way ANOVA followed by Šídák's multiple comparisons test. **P < 0.01; ****P < 0.0001. **e** Clonal expansion of CFSE-labeled OT-1 cells was analyzed in vivo after i.v. injection into PLGA-MP-vaccinated C57BL/6J mice (n = 3) (**e**) or in vitro by T-cell count after co-culture with PLGA-MP-treated BMDCs (n = 6, MP Empty, black diamonds; MP OVA/pIC, blue circles, MP OVA/Rib, red squares) or unpulsed BMDCs (Ø, black diamonds, n = 3) (**f**) by CFSE dye dilution using flow cytometry. Statistical significance was analyzed compared to the MP Empty group by one-way ANOVA followed by Tukey's multiple comparisons test. ****P < 0.0001. **g** Immune responses against indicated tumor peptide antigens were monitored by ICS for IFNγ[+] of CD8[+] splenocytes 6 days after vaccination of AAD mice (n = 3–20) with 50 µg of different TAA and 2.5 µg Riboxxim encapsulated into PLGA-MP. Mice immunized with MP Empty served as a control (Empty). All data are presented as means ± SD. Graphs represent pooled data derived from at least two independent experiments with a similar outcome.

after immunization with MP-OVA/Riboxxim, when the maximal CTL response had emerged. Tumor growth was only slightly retarded by treatment with MP-OVA/polyI:C leading to lower survival rates while MP-OVA/Riboxxim treatment significantly impeded tumor progression. Noticeably, the therapeutic potential of Riboxxim was also detectable in the absence of the tumor antigen, as MP-Riboxxim but not MP-polyI:C showed a significant tumor-suppressing effect. The superior antitumor efficacy of PLGA-MP containing Riboxxim was demonstrated by a comparative analysis of the therapeutic potential of other TLR ligands or immunostimulants displaying antitumor activity. Therefore, we encapsulated either the TLR4 ligand MPLA (monophosphoryl lipid A), the TLR7/TLR8 agonist Resiquimod (R848), or the CD1-ligand α-Galactosylceramide (α-Gal-Cer) into OVA-containing PLGA microparticles (Supplementary Fig. 9a). Despite comparable activation of antigen-specific CD8[+] T-cell responses by MP-OVA/R848 to MP-OVA/Riboxxim, a therapeutic administration induced only a mediocre antitumor response against established E.G7 tumors, which was similar to the antitumor efficacy of MP-OVA/polyI:C (Supplementary Fig. 9b, c).

Tumor immunogenicity greatly affects the susceptibility to immunotherapy and subsequently the treatment success. To evaluate the effectiveness of the PLGA-MP cancer vaccine and Riboxxim in poorly immunogenic OVA-expressing MO5 melanoma cells, mice were first immunized using a subcutaneous prime and intranasal boost vaccination scheme[24] in order to direct the antitumor response towards the lung. A PLGA-MP booster application into the tumor microenvironment seems to recruit and amplify the antitumor response by systemic and local clonal expansion of tumor-specific T cells (Supplementary Fig. 8b, c). The formation of lung metastases was significantly suppressed in the MP-OVA/Riboxxim group and to a lesser extent in mice immunized with OVA/polyI:C microparticles (Fig. 5e). Since the accumulation of tumor-specific CTLs in the tumor microenvironment positively correlates with better clinical outcomes of cancer immunotherapies[25], the infiltrates of CD8[+] cells were analyzed by immunohistochemistry. The number of CD8[+] T cells was prominently increased in the lungs of mice treated with MP-OVA/Riboxxim, while MP-OVA/polyI:C immunized mice exhibited less CD8[+] T-cell infiltration. These data suggest, that Riboxxim promotes recruitment of tumor-specific effector CD8[+] T cells to the tumor site probably due to enhanced DC function.

To assess the therapeutic efficacy of PLGA-MP antitumor therapy for already established lung metastases, mice were injected i.v. with MO5-luc[+] melanoma cells. When tumors became detectable, mice with equal tumor burden were treated with PLGA-MP via simultaneous s.c. and i.n. double route-immunization. Vaccination with MP-OVA/Riboxxim resulted in a significant reduction of tumor growth with a greater potency compared to mice treated with MP-OVA/polyI:C (Supplementary Fig. 10). This further demonstrates the antitumor potential of PLGA-MP therapy against pulmonary metastases.

Immunosuppressive molecules such as checkpoint modulators are known to dampen cytotoxic effector functions of tumor-specific CD8[+] T cells during antitumor responses. To augment MP-OVA/Riboxxim mediated tumor retardation, four accessory systemic injections of an immune checkpoint inhibitory anti-CTLA-4 monoclonal antibody (mAb) was added to the therapeutic regimen (Fig. 6a). This combination therapy significantly enhanced the suppressive efficacy of PLGA-MPs and resulted in increased survival and complete remission of E.G7 tumors in 6 out of 10 mice (Fig. 6b). By blocking of immunosuppressive factors in the tumor microenvironment, PLGA-MP mediated antitumor immunity is reconstituted leading to the eradication of large tumor masses (Fig. 6c). Remarkably, monotherapy with anti-CTLA-4 mAb did not result in suppression of tumor growth. The depletion of CD8[+] T cells during the establishment of MP-OVA/Riboxxim associated tumor suppression completely abrogated the antitumor response, indicating the indispensability of CD8[+] T cells for the effectiveness of PLGA particles as cancer vaccines. Tumor-free mice that were initially treated with MP-OVA/Riboxxim alone or in combination with anti-CTLA-4 mAb were re-challenged on day 60 post tumor challenge with E.G7 tumor cells at the contralateral side of the primary tumor site. All animals rejected re-challenged tumors, which demonstrates the generation of long-term memory (Fig. 6d). The synergistic antitumor effect of anti-CTLA-4 in our systems relies on the reinvigoration of tumor-reactive CTLs by re-enabling CD28 co-stimulation with DCs. Due to the PLGA-MP mediated depot effect, DCs are constantly activated to deliver prolonged antigen presentation and T-cell priming and thus extended antitumor CD8[+] T-cell responses. To demonstrate the significance of CTLA-4 inhibition in our vaccine system, therapeutic treatment of either anti-PD-1 or anti-CTLA-4/anti-

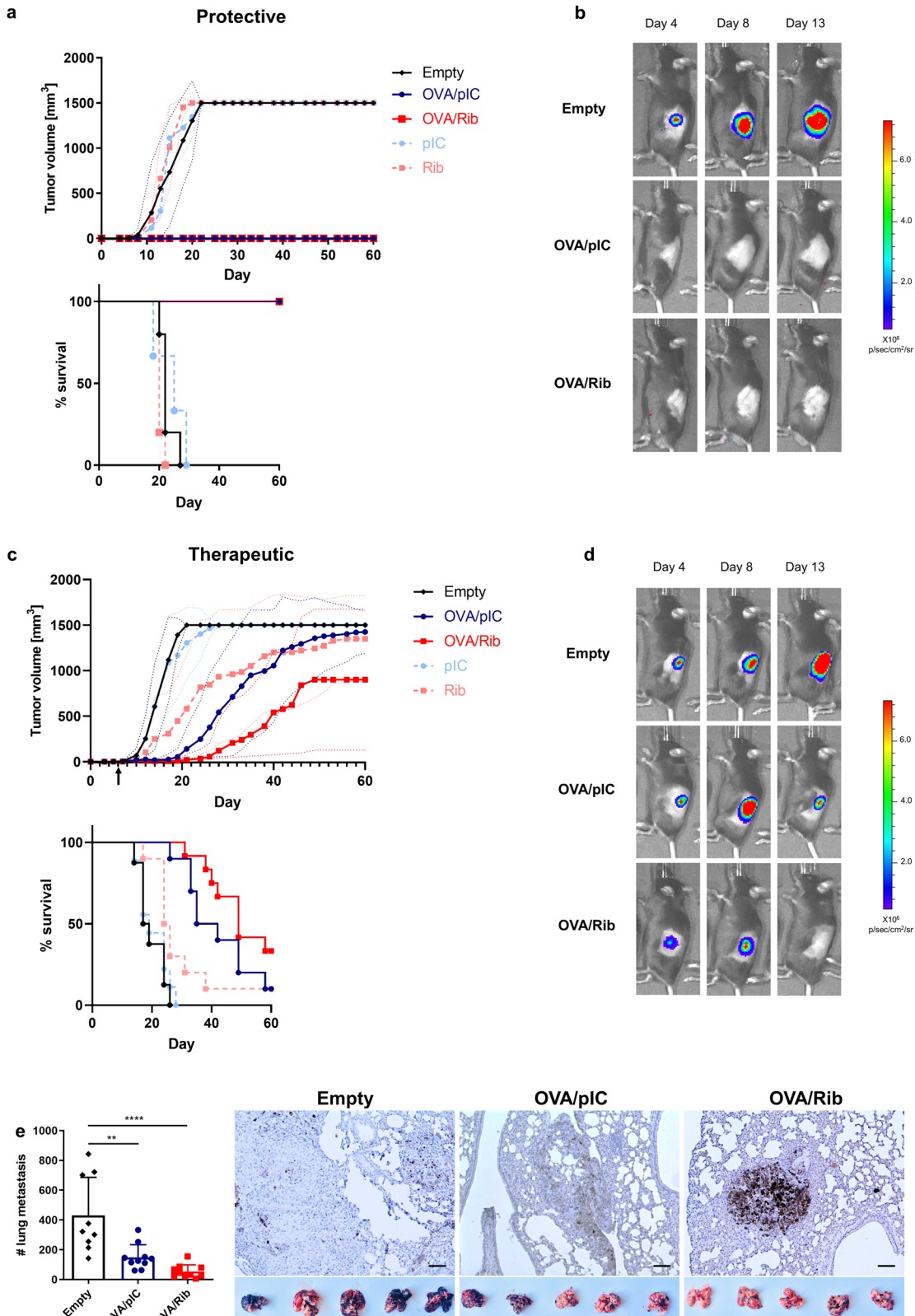

PD-1 together with PLGA microparticles was evaluated against established E.G7 tumors (Supplementary Fig. 11a, b). The combination of anti-PD-1 antibodies with PLGA particle therapy did not result in significant improvement over therapeutic tumor vaccination with MP-OVA/Riboxxim. However, the addition of a PD-1 inhibitor to the anti-CTLA-4/PLGA-MP-setup supports tumor eradication efficacy and prolongs survival. However, the synergistic effect of this combined treatment is dominated by the CTLA-4 blockade. Noticeably, monotherapy with anti-PD-1 antibody alone was ineffective, resulting in low survival rates.

The addition of immune checkpoint blockade, especially of anti-CTLA-4, greatly improves PLGA-MP-mediated cancer immunotherapy by a reinvigoration of antitumor immunity after cessation of microparticle-induced tumor-specific CTL response.

**Fig. 5 Superior immunotherapeutic efficacy of PLGA-MP containing tumor antigens and Riboxxim in multiple tumor models. a** C57BL/6J mice were immunized subcutaneously with 5 mg PLGA-MP-OVA/polyI:C (OVA/pIC, blue circles, n = 5), PLGA-MP-OVA/Riboxxim (OVA/Rib, red squares, n = 5), empty control PLGA-MP (Empty, black diamonds, n = 5) or adjuvant control MP (pIC, light blue circles, n = 3; Rib, light red squares, n = 5). Six days post-immunization, mice were s.c. inoculated with $5 \times 10^5$ E.G7-OVA-luc$^+$ thymoma cells into the left flank. Tumor growth curves and Kaplan–Meier survival analysis of the protective E.G7 tumor model are shown. Data are presented as mean ± S.D. with dotted lines in corresponding colors demonstrating individual data variances. **b** Representative IVIS® images demonstrating tumor protection after PLGA-MP antitumor treatment. Scaling of the bioluminescent pseudo-color code is shown and presented as photons/seconds/cm$^2$/steradian (p/sec/cm/sr). **c** For a therapeutic setting, C57BL/6J (n = 10) mice were s.c. challenged with $5 \times 10^5$ E.G7-OVA-luc$^+$ cells. As soon as palpable tumors were detectable, mice were immunized with PLGA-MP-OVA/polyI:C (OVA/pIC, blue circles, n = 10), MP-OVA/Riboxxim (OVA/Rib, red squares, n = 10), empty control PLGA-MP (Empty, black diamonds, n = 10) or adjuvant control MP (pIC, light blue circles, n = 9; Rib, light red squares, n = 10). Tumor growth curve and overall survival are presented showing that OVA/Rib particles significantly delayed tumor growth. Data are presented as mean ± S.D. with dotted lines in corresponding colors demonstrating individual data variances. Results are representative of two independent experiments with a similar outcome. **d** Representative IVIS® images showing tumor regression after therapeutic PLGA-MP vaccination. Scaling of the bioluminescent pseudo-color code is depicted on the right and presented as photons/seconds/cm$^2$/steradian (p/sec/cm/sr). **e** C57BL/6J mice were subcutaneously primed with MP-OVA/polyI:C (OVA/pIC, blue circles, n = 10) or MP-OVA/Riboxxim (OVA/Rib, red squares, n = 10) on day −21. Control mice were injected with the corresponding amount of empty particles (Empty, black diamonds, n = 9). 14 days later, mice received an intranasal boost with 2.5 mg of the respective microparticles (day −6) corresponding to 125 μg tumor antigen and 1.25 μg dsRNA adjuvant. On day 0, immunized mice were challenged by intravenous injection of $1 \times 10^5$ OVA-expressing MO5 melanoma cells. 21 days post tumor inoculation, numbers of melanoma tumor foci in the lungs were determined. Images (×200) representative of lung tumor tissue sections at the endpoint showing CD8$^+$ T-cell infiltration into the tumor nodule as determined by immunohistochemical staining for CD8. Scale bars, 50 μm. Representative photographs of dissected lungs from melanoma-bearing mice of the indicated treatment group are shown below. Statistics: one-way ANOVA with a Tukey's multiple comparisons test. **$P < 0.01$; ****$P < 0.0001$. Data represent means ± SD from three independent experiments with a similar outcome.

This synergy enables the eradication of large tumor masses and long-term protection from tumor relapse.

## Discussion

The importance of dsRNA agonists in cancer vaccines has been demonstrated in several studies as it facilitates potent DC activation and subsequent priming of tumor-specific CTLs with induction of a tumor-suppressive microenvironment[26]. In contrast to Riboxxim, polyI:C exhibits major disadvantages due to its undefined chemical structure of poorly annealed double and single-stranded RNA molecules with a vast size range from 1 to 8 kb[27]. As a result, the use of polyI:C in clinical trials of advanced cancer patients has entailed severe adverse effects such as hypersensitivity reactions, arthlargia, flu-like symptoms, renal failure and even failures of the cardio-vascular system in the toxic shock syndrome[28, 29]. The rapid degradation of polyI:C in body fluids[30] is another issue that results in imponderable pharmacokinetics and dynamics. In order to optimize its physicochemical properties, several approaches have been described to decrease toxicity with polyI:C derivatives. Stability in body fluids was improved by the addition of poly-L-lysine and carboxymethylcellulose to form polyICLC (Hiltonol® Oncovir)[31] while mismatched uracil and guanosine residues were added to the polyI:C structure to decrease toxicity issues in Ampligen™ (poly:IC$_{12}$U, rintatolimod)[32]. The adjuvant effect of polyIC analogs includes dose-dependent stimulation and activation of DCs expressing pro-inflammatory cytokines and type I IFNs to promote Th1 polarization and activation of antigen-specific cytotoxic T cells, with too high doses leading to inhibition of the immune activity. The additional activation of MDA5 might also lead to overshooting immune responses. Over 60 clinical studies involve the administration of the dsRNA analog Hiltonol® in solid tumors in combination with peptide vaccines, fusion proteins, or immune checkpoint blockade. However, the therapeutic effect largely depends on multiple injections using high doses of polyICLC. Repeated systemic administration of TLR agonists may induce immune hypo-responsiveness, which is commonly known as TLR tolerance[33]. Only a few clinical studies encompassed Ampligen for cancer therapy by inducing DC maturation and Th1-type T-cell immunity[34]. The multimodal immunomodulatory properties of Ampligen are not extensively characterized, which marks Ampligen as a bona fide TLR3 agonist[35].

In contrast to multiple prime-boost injections, PLGA particles carrying Riboxxim show antitumor responses after a single treatment. Compared to other dsRNA mimetics, Riboxxim displays superior properties as immune adjuvant due to the absence of systemic cytokinemia or adjuvant toxicity. The 50 bp derivative of Riboxxim, designated Riboxxol (RGIC®50), potently improves murine and human cDC1 activation to enhance T-cell proliferation[36]. The induction of a type I IFN response and apoptotic tumor cell death was enabled by targeted delivery of Riboxxol coupled to an anti-prostate stem cell antigen (PSCA) antibody derivative[36]. Matsumoto and colleagues created the chimeric TLR agonist ARNAX, which exclusively triggers TLR3 by the addition of a 5'phosphorothioate oligodeoxynucleotide (ODN)-cap of GpC DNA to the 140 bp dsRNA that specifically targets ARNAX to the endosome. ARNAX therapy-induced robust antitumor immune responses without the induction of systemic inflammation[37]. This non-inflammatory TLR3 agonist was shown to synergize with PD-L1 blockade in tumor eradication in immunocompetent hosts to overcome PD-1/PD-L1 resistance and provide long-term protection from tumor relapse. The authors highlight the necessity of both proper DC priming by a TLR3 ligand and checkpoint inhibitor blockade for efficient cancer immunotherapy[38–43].

Therapeutic targeting of RIG-I has recently emerged as another aim in antitumor immunotherapy and is currently explored in clinical trials. A combinatory vaccination scheme using 5'-tri-phosphorylated RNA-driven RIG-I stimulation with CTLA-4 or PD-1 blockade already has been demonstrated to induce potent cross-priming of CD8$^+$ CTLs and antitumor immunity in mouse models of melanoma or colorectal carcinoma[44, 45] and in human ovarian cancer[46]. The addition of 5'pppRNA to a cancer-peptide containing NP was shown to specifically boost CD8$^+$ T-cell population while not inducing suppressive immune cell types[45].

A central current challenge in cancer immunotherapy is how the striking therapeutic success of immune checkpoint blockade can be extended to patient numbers beyond 20% and to major solid tumors like prostate and breast cancer, which have not responded well to immune checkpoint blockade so far. In this study, we have shown that the combination of immune

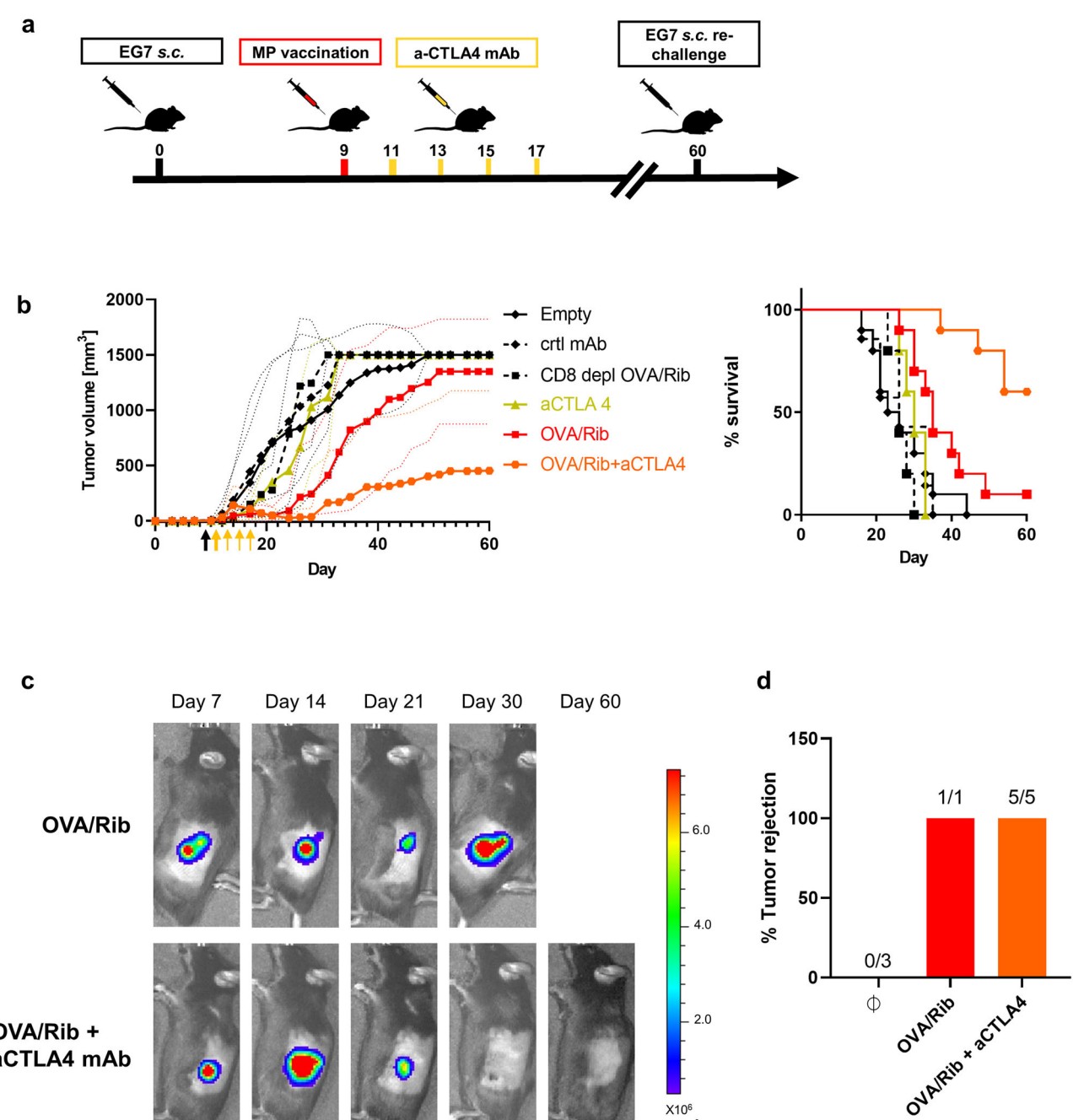

**Fig. 6 CTLA-4 blockade enhances the therapeutic effect of cancer immunotherapy using PLGA-MP enabling tumor growth retardation and prolonged survival. a** Scheme of PLGA-microparticles (MP) and immune checkpoint blockade combination therapy. **b** E.G7-OVA-luc+ subcutaneous (s.c.) tumor-bearing C57BL/6J mice were treated with PLGA-MP (250 μg OVA, 2.5 μg Riboxxim, OVA/Rib, red squares, $n = 10$), PLGA-MP (250 μg OVA, 2.5 μg Riboxxim) in combination with systemic anti-CTLA-4 (250 μg/mouse) (OVA/Rib+aCTLA4, orange hexagons, $n = 10$), or with anti-CTLA-4 monotherapy (250 μg/mouse, aCTLA4, green triangles, $n = 5$). Mice immunized with respective isotype control monoclonal antibody (mAb) (ctrl mAb, black diamonds in dotted line, $n = 5$) or MP Empty (Empty, black diamonds, n = 10) served as control. CD8+ T-cell depletion (100 μg/mouse of anti-CD8 mAb, CD8 depl OVA/Rib, black squares in dotted line, $n = 5$) was initiated 3 days before and on the day of microparticle injection and lasted until the end of the experiment with a weekly i.p. injection, demonstrating CD8+ T-cell indispensability in PLGA-MP-mediated tumor vaccination. Tumor growth is presented over time. Survival rates are expressed using Kaplan–Meier overall survival over time. Data are presented as mean ± S.D. with dotted lines in corresponding colors demonstrating individual data variances. Data were replicated in two independent experiments with a similar outcome. **c** Representative bioluminescence images of treated mice at indicated time points are shown. Scaling of the bioluminescent pseudo-color code is depicted and presented as photons/seconds/cm2/steradian (p/sec/cm/sr). **d** 60 days after tumor challenge, tumor-free mice of the MP-OVA/Riboxximm (OVA/Rib, red, $n = 1$) and the MP-OVA/Riboxxim plus anti-CTLA-4 combination therapy group (OVA/Rib + aCTLA4, orange, $n = 5$), as well as naive C57BL/6 mice ($n = 3$, Ø) were subcutaneously re-challenged with $5 \times 10^5$ E.G7-OVA-luc+ cells and tumor rejection was monitored for additional 30 days, demonstrating memory CD8+ T-cell formation causing rejection of re-implanted tumor cells.

checkpoint blockade with PLGA particle vaccine containing TLR3/RIG-I ligand Riboxxim and tumor antigen shows a synergistic efficacy in suppression of tumor growth (Fig. 6). Anti-cancer vaccination is still limited by the lack of potent GMP-certified adjuvants for use in human medication. In our study, we have focused on the pharmaceutically well-defined dsRNA adjuvant Riboxxim, which fulfills these criteria. In contrast to other TLRs, TLR3 stimulation via its signaling adapter TICAM-1 (TRIF) leads to the production of type I interferons and pro-inflammatory cytokines by downstream activation of IRF-3/7 and NF-κB, respectively. This is consistent with the observed maturation of primary mouse BMDC and primary human peripheral blood CD1c+ and CD141+ DCs as well as the induction of IFNα/β and IL-6/TNF (Fig. 3 and Supplementary Fig. 3). Potent elicitation of type I interferons by Riboxxim, which is largely attributed to activation of the RIG-I/MAVS pathway[47], promotes Th1 polarization and activation of CTLs, which we have observed in mice (Figs. 1, 4 and Supplementary Fig. 4). Moreover, IFNα/β is directly involved in cross-priming by DCs[48, 49], which will be advantageous for CTL stimulation by DC that have engulfed antigen and Riboxxim-bearing PLGA particles. The increased induction of pro-inflammatory cytokines IL-1β, IL-6, IL-12, and TNF by conventional polyI:C (Merck), which has been most widely used in preclinical studies, probably originated from pyrogen contaminations, which was markedly reduced in pyrogen-free polyI:C (Invivogen) or in GMP-grade Riboxxim (Riboxx Pharmaceuticals) (Supplementary Fig. 3b). In addition to the endosomal TLR3, polyI:C may also stimulate MDA-5 (melanoma differentiation-associated gene 5)[47, 50, 51] in an uncontrollable manner[52]. This might contribute to overshooting immune responses by activation and stimulation of several pathways. The activation of these innate immune pathways is dependent on ligand specificities and length of the dsRNA molecule. While TLR3 signaling is activated by approximately 40 to 50 bp of dsRNA, RIG-I is stimulated by dsRNA of 300–1000 bp or by short dsRNA carrying uncapped 5'triphosphate moieties and MDA-5 by more than 1000 bp[53]. Additionally, the single-stranded RNA segments in the polyI:C macromolecule also triggers TLR7/8[54] activation. Taken together, immune stimulation by the pharmaceutically well-defined 100 bp double-stranded Riboxxim is unlikely to cause adverse effects, which have been observed in clinical trials with polyI:C[28, 29]. Only the combinatory stimulation of TLR3 mediated DC activation and maturation and RIG-I induced production of type I IFNs, demonstrates the immunostimulatory and immunomodulatory potential of Riboxxim.

Concomitant delivery of tumor antigen and TLR adjuvant into the same endo-lysosomal compartment is a major advantage of cancer vaccination with PLGA-MP. Thus, it triggers proper DC maturation and improves cross-presentation to CTLs[15, 55, 56]. When comparing PLGA nanoparticles with microparticles we found that both elicited similar CTL responses in mice (Fig. 1) and that s.c. immunization yielded the strongest responses while other administration routes were immunogenic as well including i.m. inoculation, which is most frequently used for vaccine applications in humans. An extension of our study to intratumoral injection is warranted as deposition of Riboxxim-containing PLGA-MP in tumor tissue holds the potential to render the tumor micro-environment less hostile to T cells and to convert "cold" (i.e. non-inflamed) tumors into hot (i.e. inflamed) tumors, a process much facilitated by type I interferons[57]. Most of our studies were performed with the model tumor antigen ovalbumin but strong CTL responses to several well-characterized peptidic cancer antigens were obtained in AAD mice as well (Fig. 4g and Supplementary Fig. 6f). Other formulations of antigens that deserve to be systematically tested in combination with Riboxxim are long peptides,

which contain CTL and T helper cell epitopes[58] as well as lysates from excised tumor biopsies. Previously, we have shown tumor development suppression in transgenic prostate carcinoma-bearing mice using PLGA-MP containing tumor lysate[17]. The co-encapsulation of Riboxxim with tumor lysates into PLGA particles is currently tested in our laboratory, as this approach—if effective—will allow the stimulation of CTLs directed to autologous tumor-specific antigens, e.g., mutated neo-antigens that vary from tumor to tumor. By combining immune checkpoint blockade with cancer vaccination using PLGA-MP or NP containing tumor lysate and Riboxxim, "personalized" tumor-specific CTL responses might be triggered in patients who do not respond to immune checkpoint blockade alone.

In conclusion, we have characterized the immune-potentiating qualities of Riboxxim enhancing its appeal as a potent cancer vaccine adjuvant for clinical use. Due to its immunostimulatory properties in activating murine and human DC and the induction of TLR3/RIG-I mediated antitumor immunity, PLGA particle-based cancer immunotherapy with Riboxxim demonstrated great proficiency for potential use in tumor vaccination—also in combination therapy with immune checkpoint inhibition by demonstrating reactivation of PLGA-particle-elicited antitumor T cells.

## Methods

**Animals**. C57BL/6J mice and BALB/cAnNCrl (BALB/c) mice were originally purchased from Charles River (Sulzfeld, Germany). C57BL/6-Tg(TcraTcrb)1100Mjb/J (OT-1) mice and B6.Cg-Immp2l^Tg(HLA-A/H2-D)2Enge/J (AAD) mice and B6;129-Mavs^tm1Zjc/J (Mavs^−/−) mice were obtained from The Jackson Laboratory (Bar Harbor, ME, USA). B6;129S1-Tlr3^tm1Flv/J (Tlr3^−/−) mice were obtained from the Laboratory Animal Services Center (LASC)-Vivaria, University of Zurich, Switzerland. Transgenic animals are verified by genotyping of ear biopsies. Primers are listed in Supplementary Table 1. OT-1 mice were selected after flow cytometric identification of Vα2/Vβ5 staining on CD8+ T cells in blood after vena facialis puncture. Animals were further bred in the accredited animal facility of the University of Konstanz under specific pathogen-free conditions on a 12-h light/dark cycle with lights on at 7 a.m. Mice were kept in air-conditioned rooms with controlled temperature (22 °C), 55% relative humidity, and constant ventilation (17 air changes/h). Animals were provided ad libitum access to standard, autoclaved laboratory animal diet, and tap water. Male and female mice were used at 8–12 weeks of age. The number of animals in each group was determined according to statistical verification and previous studies.

**Ethics statement**. Animal experiments have been conducted in compliance with ethical standards of German and EU guidelines after approval by the animal experimentation ethics committee of the Review Board of Governmental Presidium Freiburg, Germany (approval numbers G-15/102, G-16/81, G-19/176, G-20/03, G-20/143). Human specimens were used in accordance with the requirements of the institutional ethics committee and national policies. Blood donations were approved by the Ethics Committee of the University of Konstanz, Germany. Written informed consent was obtained from each of the randomly enrolled healthy volunteers before blood donation following guidelines of the Review Board protocol of the University of Konstanz. Blood donations were performed by the transfusion unit at the Klinikum Konstanz in accordance with the precepts of the transfusion unit at the hospital and of the DRK (German Red Cross) with all applicable regulations, guidance, and local medical policies.

**Cell lines and culture media**. The murine lymphoma cell line stably expressing OVA and Luciferase, E.G7-OVA-luc+ cells were kindly provided by Prof. Thomas Blankenstein (MDC Berlin, Germany) and maintained in RPMI 1640 plus 10% (v/v) FCS, 100 IU/ml Penicillin and 100 μg/ml Streptomycin. MO5 cells, murine melanoma cells stably expressing OVA, were provided by Dr. Antje Heit (TU Munich, Germany) and cultured in DMEM supplemented with 10% (v/v) FCS, 100 IU/ml Penicillin, and 100 μg/ml Streptomycin. Luciferase-positive, OVA-expressing B16BL6 cells were generated by lentiviral transduction of cytosolic ovalbumin (amino acids 51-386) and cultured in DMEM and 10% (v/v) FCS, 100 IU/ml Penicillin, and 100 μg/ml Streptomycin. The original cell line B16BL6-luc+/GFP+ was kindly provided by Prof. Olaf van Tellingen (The Netherlands Cancer Institute, Amsterdam, The Netherlands). To maintain Luciferase expression, respective growth media were supplemented with 0.4 mg/ml G418 Sulfate (Geneticin™, Gibco, Thermo Fisher Scientific). Positive expression of luciferase was verified before every injection into animals by in vitro bioluminescence imaging on the IVIS® Spectrum 200 platform (Perkin Elmer). Expression of ovalbumin was authenticated via immunoblotting using anti-chicken egg albumin antibody (#C6534, Merck). The immature dendritic cell line DC2.4 (#SCC142, Merck) was provided by Kenneth

Rock (Worcester, MA, USA) and maintained in RPMI 1640. The CD8$^+$ splenic DC cell line MutuDC2114[59] was a kind gift of Prof. Hans Acha-Orbea (University of Lausanne, Switzerland) and cultured in IMDM with 10% (v/v) FCS, 100 IU/ml Penicillin, and 100 μg/ml Streptomycin. The B3Z T-cell hybridoma[60] was a kind gift from Prof. Nilabh Shastri (University of California, Berkeley, USA) and cultured in IMDM supplemented with 10% (v/v) FCS, 100 IU/ml Penicillin, and 100 μM Streptomycin. The CD4$^+$ T-cell hybridoma DOBW[61] was kindly contributed by Prof. Clifford V. Harding (Washington University, School of Medicine, St. Louis, USA) and maintained in DMEM supplemented with 10% (v/v) FCS, 100 IU/ml Penicillin and 100 μg/ml Streptomycin, 1 mM sodium pyruvate, 10 mM HEPES and 0.5 mM β-mercaptoethanol. The human monocytic cell line THP-1 (#TIB-202™) was purchased by ATCC® (Virginia, USA) and maintained in RPMI 1640. All culture media were supplemented with GlutaMAX™. Media and standard additives were purchased from Gibco® (ThermoFisher Scientific). Cell lines were regularly tested negative for *Mycoplasma* by PCR analysis of cell culture supernatants by Microsynth, Switzerland. Cell cultures were maintained at 37 °C with 5% CO$_2$ in a humidified atmosphere. Adherent cells were detached by incubation with 0.05% trypsin-EDTA for 5 min at 37 °C. Primary DCs were detached by 5 mM EDTA/PBS for 15 min at 37 °C.

**Preparation and characterization of PLGA particles.** PLGA particles were prepared from Resomer RG502H (Evonik Industries). Nanoparticles (NP) were produced by the double emulsification solvent evaporation method as previously described[62]. 5 mg ovalbumin (albumin from chicken egg white, #A251, Merck) and 0.05 mg adjuvant in 100 μl 0.1 M NaHCO$_3$ were dissolved in the primary emulsion (100 mg PLGA in 3 ml dichloromethane (DCM)) and mixed with 25 ml of 2% poly (vinyl alcohol) (PVA, MW 9000–10,000, #360627, Merck). The mixture was stirred overnight at 4 °C to evaporate DCM. NPs were collected by ultra-centrifugation and lyophilized at −80 °C for 48 h. PLGA microparticles (MP) were prepared by spray-drying. For antigen/TLR3L co-encapsulation, 50 mg OVA or 10 mg of peptide antigens and 0.5 mg adjuvants (polyI:C sodium salt, Merck; poly(I:C) (HMW), InvivoGen; Riboxxim™, Riboxx Pharmaceuticals; MPLA-SM VacciGrade™, Invivo-Gen; R848 VacciGrade™, InvivoGen; α-Galactosylceramide (α-Gal-Cer, KRN700, Funakoshi, Japan)) were dissolved in 1 ml of 0.1 M NaHCO$_3$ (aqueous phase) and emulsified with 1 g PLGA in 20 ml dichloromethane (organic phase) using a digital microtip sonicator. Riboxxim is provided at a 1 mg/ml solution in water for injection and is certified as GMP material sterile and endotoxin-free (European guidelines ICH-6). The HLA-A*0201 restricted tumor-associated peptides STEAP1$_{262–270}$ (LLLGTIHAL); STEAP1$_{292–300}$ (MIAVFLPIV); PAP$_{135–143}$ (ILLWQPIPV); PSMA$_{469–477}$ (LMYSLVHNL); Her2$_{689–697}$ (RLLQETELV); TRP-2$_{180–188}$ (SVYDFFVWL) and NY-ESO1$_{157–165}$ (SLLMWITQV) were synthesized by the Institute for Biochemistry, Charité Berlin or were purchased by peptides&elephants, Germany. The obtained dispersion was immediately spray-dried with the Mini Spray-Dryer 290 (Büchi Labortechnik AG) at a flow rate of 1 ml/min and inlet/outlet temperatures of 25 °C/23 °C. Spray-dried microparticles were washed out of the spray-dryer's cyclone with 0.05% Poloxamer 188 (#P5556, Merck) and collected on a cellulose acetate membrane filter. PLGA-MP were dried under vacuum at room temperature and subsequently stored under desiccation at 4 °C.

The amounts of encapsulated antigens and TLR ligands were determined after dissolving the PLGA particles in DMSO for 10 min. Phase separation of precipitated polymer and soluble compounds was achieved by the addition of 0.1 M NaOH. Antigen release was quantified using Pierce™ MicroBCA Protein Assay Kit (Thermo Fisher Scientific) after incubating particles in PBS pH 7.4 at 37 °C under continuous orbital agitation.

The particle size and polydispersity index (PDI) were analyzed by cumulative analysis of dynamic light scattering (DLS) using the Zetasizer® Nano ZSP particle size analyzer and Zetasizer software version 7.12 (Malvern Panalytical). The ζ (zeta) potential of particles was measured by laser Doppler velocimetry in combination with phase analysis light scattering using the same device. Particles were dispersed in ultrapure water prior to measurement at 25 °C with the Helmholtz–Smoluchowski model.

To characterize surface morphology using scanning electron microscopy (SEM) particles were mounted on aluminum stubs and sputter-coated with a 6 nm thick layer of platinum by an argon beam sputter coater (Quorum Q150R, Quorum Technologies, Lewes, UK). Micrographs were produced on a Zeiss Auriga 40 at 1, 3, or 5 kV regarding the sample with SmartSEM v6.0 (Zeiss, Oberkochen, Germany).

Endotoxin levels of PLGA particles were detected by ToxinSensor™ Chromogenic LAL Endotoxin Assay according to the manufacturer's instructions (#L00350, Genscript).

In vitro particle internalization by primary DCs and cell lines was determined using fluorescently labeled particles (PLGA-MP-OVA/Riboxxim/QD) (Qtracker™ 705 Vascular Labels, Invitrogen™, Thermo Fisher Scientific). QD-loaded MPs were added to cells at a concentration of 10 μg/ml per 1 × 10$^6$ cells/well. Particle uptake was performed for 6 h at 37 °C or at 4 °C as a control (no uptake). Phase-contrast microscopy was performed for semi-quantitative estimation of particle uptake. After incubation, cells were harvested and incubated for 1–3 s in pure DMSO to remove unbound particles. After that, cells were washed three times with 1× DPBS. QD-positive fluorescent signals were assessed for each sample by flow cytometry.

**Generation of bone marrow-derived DCs (BMDCs) and peritoneal macrophages.** DCs from proliferating mouse bone marrow progenitors were isolated from femur and tibia of 8–10-weeks-old wild-type C57BL/6J mice and differentiated according to Lutz et al.[63]. Isolated bone marrow cells were cultured for 9 days in 10 cm dishes at 2 × 10$^6$ cells in RPMI 1640, supplemented with 10% (v/v) FCS, 100 IU/ml Penicillin, 100 μg/ml Streptomycin, 1 mM sodium pyruvate, 55 μM β-mercaptoethanol and 20 ng/ml recombinant murine GM-CSF (#315-03, Pepro-Tech). Peritoneal macrophages were elicited by i.p. injection of 0.5 ml of 3% Brewer thioglycolate medium (w/v)[64]. Cells were harvested on day 3 after injection by washing cells from the peritoneal cavity by lavage with 5 ml of ice-cold, sterile PBS/3%FCS. After a centrifugation step at 300 × g for 5 min, cells were seeded at 1 × 10$^6$ cells/well in RPMI 1640, supplemented with 10% (v/v) FCS, 100 IU/ml Penicillin, 100 μg/ml Streptomycin, 55 μM β-mercaptoethanol. Non-adherent cells were discarded after 6 h. Adherent macrophages were used after overnight cultivation.

**Antibodies and flow cytometry.** Antibodies and dilutions used in this study are listed in Supplementary Table 2. Antibody staining was performed in the presence of purified CD16/CD32 antibody (Mouse BD Fc-Block™, clone: 2.4G2, #553141, BD Biosciences) at a concentration of 1 μg per 1 × 10$^6$ cells. If required, cells were stained after RBC-lysis with 1.66% NH$_4$Cl w/v. Live/Dead cell discrimination was performed by the addition of Zombie Green™ Fixable Viability Dye (#423111, BioLegend). Flow cytometry was performed using BD LSRFortessa™ or BD FAC-SLyric™ instruments. Data were acquired using BD FACSDiva™ Version 8.0.1 and BD FACSuite™ 1.2.1 (BD Biosciences). Data were analyzed using FlowJo™ v10.7.1 software (BD Biosciences).

**Differentiation and isolation of human myeloid DCs.** The human monocytic leukemia cell line THP-1 was differentiated into immature DC with similar characteristics to human monocyte-derived DCs (MoDCs) by culturing 2 × 10$^5$ THP-1/ml in the presence of rhGM-CSF (20 ng/ml) and rhIL-4 (20 ng/ml) for 5 days[65]. THP-1 cells were differentiated into monocyte-derived macrophages (MoMØ) using 100 nM of phorbol 12-myristate 13-acetate (PMA, #P1585, Merck) for 72h[66]. Blood samples from healthy donors were collected as whole blood donation in 500 ml blood bags supplemented with CPDA-1 stabilizer. Human peripheral blood mononuclear cells (PBMC) were isolated within the first hour of acquisition by Ficoll®Paque™ PLUS (GE Healthcare, #17144002) centrifugation of 35 ml fresh blood in 50 ml LeucoSep tubes (Greiner) at 800×g for 15 min at RT without brake. Mononuclear cells at the interphase were purified by multiple washing steps with 1× DPBS. Human primary cells were cultured in AIM-V medium (Gibco, Thermo Fisher Scientific) supplemented with 2% heat-inactivated human AB serum (#HUAB.SE, BioSell) and 50 μM β-mercaptoethanol (complete AIM-V). CD14$^+$ monocyte-derived DCs (MoDCs) were enriched by positive magnetic selection using anti-CD14 microbeads according to the manufacturer's instructions (#130-050-201, Miltenyi Biotec). Purified CD14$^+$ monocytes were differentiated into MoDCs by culturing 2 × 10$^6$ cells/ml in complete AIM-V medium supplemented with 500 U/ml recombinant human GM-CSF (#300-03, PeproTech) and 250 U/ml recombinant human IL-4 (#200-04, PeproTech) for 5 days. CD14$^+$ monocytes were differentiated into CD68 + MoMØ by culturing 2 × 10$^6$ cells/ml in a complete AIM-V medium supplemented with 50 ng/ml rhGM-CSF for 5 days[67]. CD1c$^+$ (BDCA-1$^+$) and CD141$^+$ (BDCA-3$^+$) myeloid DCs were purified from CD14-negative PBMCs using MACS isolation kits (#130-119-475, Miltenyi Biotec) or magnetic MicroBeads (#130-090-512, Miltenyi Biotech), respectively. CD1c$^+$ and CD141$^+$ cells were used directly.

**Antigen-specific T-cell response.** Intracellular cytokine staining (ICS) was performed to detect intracellular IFNγ as a measure of cytotoxic T lymphocyte activation. Splenocytes or tumor-infiltrating lymphocytes (TILs) were cultured with or without 10$^{−6}$ M SIINFEKL peptide for 5 h in the presence of 10 μM/ml Brefeldin A (#B6542, Merck). For surface staining, cells were incubated with anti-CD8α APC (clone 53-6.7, #17-0081-82, eBioscience™) for 30 min at 4 °C; then cells were fixed and permeabilized using a Cytofix/Cytoperm Kit (#554722, BD Biosciences) according to the manufacturer's instructions as to preparation for intracellular staining using anti-IFNγ BV421 antibody (clone XGM1.2, #505829, BioLegend) in BD Perm/Wash™ buffer (#554723, BD Biosciences). CD8$^+$IFNγ$^+$ cells were analyzed 16 h later by flow cytometry.

**ELISpot assay.** IFNγ production of OVA-specific cells was analyzed using a commercially available mouse ELISPOT antibody pair according to the manufacturer's protocol (#551881, BD Biosciences). Briefly, splenocytes of immunized mice were seeded at 5 × 10$^5$ cells per well onto pre-coated MultiScreen®HTS Filter Plates (Merck Millipore) and re-stimulated with 10 μM of the respective peptide for 20 h. After incubation with the biotinylated secondary antibody specific for IFNγ, a streptavidin-alkaline phosphatase enzyme conjugate was added. After the addition of the BCIP®/NBT substrate solution (#1911, Merck), a purple precipitate is formed as spots at the sites of captured IFNγ. Automated spot analysis and quantification were performed using the ImmunoScan® analyzer and Immunospot software v.6.0.0.2 (CTL Europe, Germany).

**Silver staining**. Equal amounts of supernatants of PLGA-MP release assays were resolved in reducing conditions on 10% SDS-polyacrylamide gels for electrophoresis. Silver Staining was performed according to the manufacturer's instructions (Pierce™ Silver Staining Kit, Thermo Fisher Scientific).

**Analysis of cytokine expression**. Supernatants of DC cultures and BMDC-DC co-cultures were analyzed for different cytokines by ELISA according to the manufacturers' instructions. Commercially available ELISA Kits were purchased from Thermo Fisher Scientific (Invitrogen™ eBioscience™ ELISA Ready-SET-Go!™) except for mIFNγ (ELISA MAX™ Deluxe Set Mouse IFN-γ, #430804, BioLegend) and mIFNβ (LEGEND MAX™ Mouse IFN-β ELISA Kit, #439407, BioLegend). Mouse IFNα, as well as human IFNα and IFNβ were purchased from PBL (VeriKine Mouse Interferon Alpha ELISA Kit #42120, VeriKine-HS Human Interferon Alpha All Subtype TCM ELISA Kit #41135, and VeriKine Human Interferon Beta ELISA Kit #41410, respectively, PBL Assay Science, USA).

**In vitro antigen-presentation assay**. Antigen (cross-) presentation of SIINFEKL peptide was assessed using the B3Z $CD8^+$ T-cell hybridoma, specific for the SIINFEKL peptide of ovalbumin ($OVA_{257-264}$) and the $CD4^+$ T-cell hybridoma DOBW specific for the ovalbumin class II epitope ISQAVHAAHAEINEAGR ($OVA_{323-339}$). $1 \times 10^5$ hybridoma cells were co-cultured overnight with $2 \times 10^5$ DCs loaded with PLGA particles. To determine the relative maximum of T-cell activation, DCs were pulsed with SIINFEKL peptide (final concentration of $10^{-6}$ M). 100 µl of *lacZ* buffer (0.13% NP40, 9 mM $MgCl_2$, 0.15 mM chlorophenolred-β-D-galactoside (CPRG)) was added for an additional 4-h incubation step[68]. The absorbance of the chromogenic reaction was measured at 570 nm with background subtraction at 620 nm using a spectrofluorometer (Infinite Pro 200, with Magellan Software, Tecan). Presentation on MHC class II was similarly assessed as described above, except for using the $OVA_{323-339}$-specific hybridoma cell line DOBW. IL-2 secretion was measured from supernatants of DOBW-DC co-cultures using a mouse IL-2 ELISA kit (#BMS601, Invitrogen™, Thermo Fisher Scientific) according to the manufacturer's protocol.

**Analysis of T-cell proliferation**. $CD8^+$ T cells were isolated from OT-I spleens and LN using mouse CD8a (Ly-2) MicroBeads (#130-117-044, Miltenyi Biotec), and then labeled with 1 µM of the proliferation and cell tracking dye CFSE (#423801, BioLegend). $1 \times 10^5$ labeled $CD8^+$ cells were co-cultured with an equal number of PLGA particle loaded DCs or splenocytes of PLGA-MP immunized mice in complete medium. After 72 h, cells were recovered, and OT-I proliferation was measured by CFSE dye dilution of $CD8^+$ T cells using flow cytometry. Additionally, IFNγ secretion of culture supernatants was measured by mouse IFNγ ELISA kit (ELISA MAX™ Deluxe Set Mouse IFN-γ, #430804, BioLegend) according to the manufacturer's protocol.

**In vivo cytotoxicity assay**. Cytotoxic activity of OVA-specific CTLs in PLGA-MP immunized mice were analyzed in vivo[69]. On day 5 post-vaccination, splenocytes from congenic C57BL/6J mice were pulsed with 1 µM SIINFEKL peptide (vaccine-specific target cells) or left unpulsed (non-specific control cells). Immunized mice were i.v. injected with a 1:1 mixture of $10^7$ cells labeled with either 5 µM CFSE (#423801, BioLegend) (peptide-pulsed) or 0.5 µM CFSE (unpulsed). The following day, ratios of the CFSE bright to dim fluorescent peaks of $CD45^+$ splenocytes were analyzed. OVA-specific cytotoxicity was calculated using the following equation: {1 $-$(PLGA-MP immunized mice) [$CFSE^{high}$(%)/$CFSE^{low}$(%)]/naive mice [$CFSE^{high}$ (%)/$CFSE^{low}$(%)]} $\times 100$

**In vitro cytotoxicity assay**. The viability of murine BMDCs was determined by incubation of DCs with varying concentrations of PLGA-MP in 96-well plates at $5 \times 10^4$ cells per well for 48 h at 37 °C. The number of live (metabolically active) cells was evaluated by the addition of 10% of Deep Blue Cell Viability™ Kit (#424701, BioLegend) according to the manufacturer's protocol. The reduction of resazurin into resorufin was photometrically detected using a spectrofluorometer (Infinite Pro 200, Tecan) at 550 nm. To demonstrate cytotoxicity, cells were treated with 10% (w/v) Triton®X-100 (Merck) for the last 4 h as a positive control.

**Immunization of animals with PLGA particles**. For immunization studies, 8-to 12-week-old female and male C57BL/6J, $Mavs^{-/-}$, $Tlr3^{-/-}$ or AAD mice were immunized either subcutaneously at the base of the tail or via the intraperitoneal route using 5 mg PLGA particles loaded with antigen (250 µg OVA/mouse, 50 µg peptide/mouse) and dsRNA analogs (2.5 µg/mouse) in PBS. The following applications were performed under isoflurane anesthesia. For intranasal immunizations, mice were vaccinated with a suspension of 2.5 mg PLGA particles (125 µg OVA/mouse; 1.25 µg dsRNA analog/mouse) in 50 µl PBS (25 µl per nostril). 50 µl of a 1 mg/ml PLGA particle solution in PBS (2.5 µg OVA/mouse; 0.125 µg dsRNA analog/mouse) were administered intramuscularly into the hind limb thigh muscle. The intranodal injection was also performed by administering 10 µl of a 1 mg/ml PLGA particle suspension in PBS (0.5 µg OVA/mouse; 0.025 µg dsRNA analog/mouse) into the inguinal lymph node. Control mice were treated with respective amounts of empty PLGA particles or particles containing only dsRNA analogs. On

day 6 after immunization, target organs were resected and homogenized into single-cell suspension by smashing through 70 µm cell strainers (BD Falcon) for further analysis. Lung tissue was dissociated using the mouse Lung Dissociation Kit (#130-095-927, Miltenyi Biotec) with a gentleMACS™ octo Dissociater according to the manufacturer's protocol.

**In vivo study of the distribution of QD-positive PLGA-MP**. Female BALB/c mice were s.c. injected with 5 mg of QuantumDot containing PLGA-MP-OVA/Riboxxim to the left flank. The left non-injected flank served as a negative control. Fluorescence signals were acquired at indicated time points with the IVIS® Spectrum 200 In Vivo Imaging System (Perkin Elmer) using the Far-red filter set. Popliteal and inguinal LNs were dissected at different time points post-immunization and incubated in 10% neutral formalin for subsequent paraffin-embedding and histology.

**Histology and immunohistochemistry (IHC)**. Organs were excised and immediately fixed in 10% neutral-buffered formalin for 24 h followed by paraffin-embedding and sectioning to 5-µm-thick slides. H&E staining was performed according to standard protocols. For IHC staining, tumor-bearing tissues were collected and processed as described before. Paraffin-embedded tissue sections of 3 µm thickness were rehydrated and subjected to heat-induced antigen retrieval (HIER) in 10 mM citrate buffer (pH 6). Blocking was performed with 10% goat serum. Sections were incubated with rat primary antibodies specific for mouse CD8a (1:200) (clone D4W2Z, #98941, Cell Signaling) at 4 °C overnight and then stained with an avidin-biotin-based peroxidase system (#PK-6100, VECTASTAIN® Elite ABC-HRP Kit, Vector Laboratories). Positive immunoreactions were visualized using peroxidase substrate kits (ImmPACT®, Vector Laboratories). Sections were counterstained with hematoxylin. Results of three random fields per section were recorded with AxioObserver or AxioImager microscopes and ZEN 3.2 (blue edition) imaging software (Carl Zeiss Imaging Inc.) and analyzed in a blinded manner.

**Tumor challenge, immune checkpoint blockade combination therapy, and bioluminescent imaging**. C57BL/6J mice were randomized into different groups ($n = 5$). The establishment of lung metastases was performed by i.v. injection of $10^5$ MO5 melanoma cells. To direct antitumor responses to the site of tumor formation, mice were immunized with PLGA particles in a prime-boost setting as previously described[24]. To this aim, mice received a subcutaneous prime immunization followed by an intranasal boost 14 d later. Therapeutic PLGA-MP treatment was performed by simultaneous s.c. and i.n. double re-immunization. Solid tumors were established by s.c. injection of $5 \times 10^5$ syngeneic E.G7-OVA-luc + cells into the left flank of each mouse. Tumor volumes were measured at regular intervals by using bioluminescence and caliper measurements of two perpendicular tumor dimensions. The tumor volume was calculated using the following formula: Tumor volume ($mm^3$) = [(long diameter) × (short diameter)$^2$] × 0.5. A single dose of PLGA-MP treatment was administered either in a protective manner 6 days before tumor challenge or therapeutically when tumor nodules became palpable. To deplete $CD8^+$ T cells, mice were i.p. injected with 100 µg of InVivoMAb anti-mouse CD8β/Lyt 2.1 mAb (clone 116-13.1 (HB129), #BE0118, BioXCell) 3 days before and on the day of tumor cell inoculation followed by weekly injections until the end of the experiment. Depletion of $CD8^+$ T cells in peripheral blood was validated by flow cytometric analysis with 100% loss of $CD8^+$ T cell at the peak of PLGA-MP-mediated antitumor CTL response (day 6 post-vaccination). Immune checkpoint blockade was achieved by multiple i.p. injections of GoInVivo™ purified anti-mouse CD152 (clone 9H10, #106205, BioLegend) with a cumulative dose of 250 µg anti-CTLA-4 per mouse or a total of 1500 µg purified anti-PD-1 mAb per mouse (RMPI-14, provided by Prof. H. Yagita (Juntendo University School of Medicine, Tokyo)). Detailed treatment regimens of immune checkpoint blockade and PLGA-MP vaccinations are outlined in schemes and figure legends. The tumor size was measured every 3 days using a caliper and bioluminescent signals were acquired at indicated time points using the IVIS® 200 imaging system (Perkin Elmer). Mice were anesthetized by isoflurane inhalation and D-Luciferin*K (#bc219, Synchem) was administered intraperitoneally at 150 mg/kg prior to luminescence imaging. Luminescent signals were acquired using a 20 cm field of view, small binning, and 120 s exposure time. Regions of interest (ROI) were preset around the signal on pseudo-color luminescent images using Living Image® v4.1 Software (Perkin Elmer). The emitted signal intensity (photon flux) was integrated over these ROIs and it is expressed as relative light units (RLU, photons per second per $cm^2$) with the lower signal threshold set to 5% of the maximum signal value. Body weights were monitored three times a week. Tumor-bearing mice were euthanized in case s.c. tumors exceeded 1500 $mm^3$ or exhibit necrotic or ulcerative pathologies and when animals showed signs of distress or rapid weight loss of 20% of the initial weight.

**Statistical analysis**. Statistical significance was determined by applying a two-tailed Student's $t$-test for two-group comparisons. One-way or two-way analysis of variances (ANOVA) followed by Tukey's or Sidak's post-hoc tests was used for comparison of multiple groups. The Kaplan–Meier survival analysis was used to estimate statistical significance in overall survival distribution between the groups and Log-rank (Mantel–Cox) tests were applied to compare survival rates. Unless

otherwise stated in the figure legends, all data were derived from at least three independent experiments and are presented as means ± standard deviation (SD). Statistical analyses were performed using GraphPad Prism 8.4.1 (GraphPad Software, Inc.). Statistical significance was achieved when $P < 0.05$.

**Reporting summary**. Further information on research design is available in the Nature Research Reporting Summary linked to this article.

## Data availability

All relevant data supporting the findings of this study are included within the article and supplementary information files or are available from the authors upon reasonable request. Source data are provided with this paper.

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

## Acknowledgements

We would like to thank Aurélie Demont (Büchi Labortechnik AG) for helpful advice for spray drying. The facilities FlowKon, BioImaging Center (BIC), Particle Analysis Center (PAC), Electron Microscopy Center (EMC) and in particular Michael Laumann, and Animal Research Center at the University of Konstanz are acknowledged for help and guidance. We are grateful to Oya Tagit and Yusuf Dölen for their introduction into NP production and intranodal injection technique. We thank Thomas Brunner for providing access to histology equipment, and Thomas Blankenstein, Antje Heit, Olaf van Tellingen, Kenneth Rock, Hans Acha-Orbea, Nilabh Shastri, and Clifford Harding for the contribution of cell lines. This project has received funding from the European Union's Horizon 2020 research and innovation program under grant agreement No. 686089, from Deutsche Krebshilfe (Grant Nr. 70112413), from Deutsche Forschungsgemeinschaft (DFG, German Research Foundation) under Germany's Excellence Strategy – EXC 2117 – 422037984, from the Novartis Foundation for medical-biological research (grant Nr. 20B131), and from Fondazione San Salvatore.

## Author contributions

J.K., D.H., and V.L.H. conceived and performed experiments and analyzed data. J.K. wrote the manuscript. D.H. prepared the figures. A.M. generated MO5-luc+ cells. B.G. provided advice and help for particle production. H.Y. contributed PD-1 antibody and J.R. provided Riboxxim and contributed to data discussion and manuscript refinement. M.G. supervised the project, discussed and analyzed data, and refined the manuscript.

## Funding

## Competing interests

J.R. is an employee of Riboxx GmbH (Germany). The remaining authors declare no competing interests.
