## [Peer Review File · Nature Communications]

REVIEWER COMMENTS

Reviewer #1 (Remarks to the Author):

The authors present a PLGA-microparticles (MPs)-based cancer vaccine carrying TLR3 agonist Riboxsim and antigens. PLGA-nanoparticles (NPs) or -MPs-based cancer vaccines containing antigens and TLR ligands have been extensively studied by many researchers including the authors. In this manuscript, the authors use Riboxsim, a recently developed TLR3 agonist, and GMP grade PLGA-MPs for possible application to clinical studies. Subcutaneous injection of PLGA-MP-OVA/Riboxsim into mice induced OVA-specific CD8+ T cell activation, and retarded tumor growth in EG7-bearing mice. Mouse and human myeloid DCs efficiently take up MP-OVA/Riboxsim. PLGA-MPs containing Riboxsim and HLA-A0201-restricted cancer peptides induced peptide-specific CD8+ T cell activation in HLA-A0201-transgenic mice. Finally, combination therapy of MP-OVA/Riboxsim and anti-CTLA4 Ab strongly induced tumor regression and prolonged survival in EG7-bearing mice. Overall, these experiments are well-designed and proceeded carefully. However, several important issues remain unsolved.

Major comments

1. The authors mention that Riboxsim is a TLR3 ligand that induces potent DC activation and secretion of type I IFNs (Fig. 3). It is surprising that extracellularly added Riboxsim (100bp dsRNA) strongly induces type I IFN production by BMDCs as compared to polyIC. Riboxsim is synthesized with DNA/RNA synthesizer or in vitro transcribed? Original paper of Riboxsim (RGC100) by Naumann et al. only shows the data from TLR3 knockdown experiments in JAWSII DCs in terms of TLR3 specificity. To confirm TLR3 specificity of Riboxsim, the authors should examine the cytokine profiles of Riboxsim-stimulated BMDC and CD8alpha+ DCs from TLR3 KO and MAVS KO mice.
2. Several reports have demonstrated that PLGA-NPs are taken up by DCs/macrophages and delivered into both endosomes and cytoplasm where encapsulated molecules are released (Reviewed by Hamdy et al., ADDR, 2011), suggesting that Riboxsim incorporated into PLGA-MPs is transferred to both endosomes and cytosol together with antigens. Thus, it is unknown whether MP-OVA/Riboxsim-mediated CD8+ T cell activation depends on endosomal TLR3. CD8+ T cell activation as well as serum cytokine levels following MP-OVA/Riboxsim injection (s.c.) should be investigated using TLR3 KO and MAVS KO mice.
3. The authors investigated the protective role of MP-OVA/Riboxsim in tumor development using EG7 and MO5 tumor models, but its therapeutic role was examined only in EG7 tumors. In cancer immunotherapy using vaccine adjuvants, therapeutic effect is more important than protective effect. Recent papers by Takeda et al., demonstrate that vaccine immunotherapy with TLR3-specific ligand ARNAX and tumor antigens induces anti-tumor CTLs and enhances anti-tumor responses in conjunction with PD-L1 blockade (Takeda et al., 2017, Cell Repos 19:1874-1887; Takeda et al., Cancer Sci. 2018, 109:2119-2129). Does therapeutic vaccination of MP-OVA/Riboxsim suppress lung metastasis of MO5? These experiments should be done. Also, citation of these articles and discussions are required.

Minor comments

1. Fig. 5c and Fig. 6b. Tumor growth data of individual mice should be shown in supplemental figures.
2. Discussion. Among the TLR3 agonists, the authors compare polyICLC and Riboxsim in terms of chemical structure, stability and adverse effects ; polyIC is heterogenous undefined dsRNA (1-8kb) while Riboxsim is 100bp dsRNA. It is wonder that the authors do not mention other TLR3 agonist such as ARNAX that has defined chemical structure and anti-tumor

efficacy via activation of professional antigen-presenting DCs (Nat Commun. 6:6280, 2015, Trial Wach, Oncoimmunology, 9:1, 2020). The authors should refer these papers.

Reviewer #2 (Remarks to the Author):

The current manuscript by Groettrup and colleagues reports on the formulation of a cancer vaccine by combining antigens with a TLR3 agonist Riboxsim in PLGA particles. The vaccine was tested in vitro and in vivo, and the results show that its vaccination efficiency in mice was significantly improved compared to that of a control formulation with Poly I:C as the adjuvant. Furthermore, the Riboxsim-based vaccine in combination with CTLA-4 extended the survival of tumor-bearing mice beyond monotherapies with either agent.

Major issues:

1. A major issue of the study is its novelty. Therapeutic or protective cancer vaccines based on adjuvants and antigens in nano/microparticles have been reported in many previous studies, with a high variety of different types of adjuvants and the antigens. The first reports on this topic can be dated back to a few decades ago. Recently there have been a considerable number of publications on combinations of cancer vaccines with other immunotherapeutics such as checkpoint blockade drugs. Besides these previous studies, the team has also published on vaccines containing their own adjuvant Riboxsim. Therefore, the current study is a useful and informative report on the vaccine developed by the authors, but it brings limited new perspectives to the research field.
2. The design of the current study is non-ideal. One major limitation is that the authors only compared their adjuvant Riboxsim to Poly I:C, which is a known standard TLR3 agonist with limited applicability due to its high toxicity. To really demonstrate the high potential of their adjuvant/vaccine regarding efficacy and safety, the study should involve other TLR agonists or even other types of vaccine adjuvants. There are quite a few vaccine adjuvants on the market, which are not based on TLR agonists. In my mind, those are important controls for this study.
3. In the combination therapy study, the vaccine was combined with CTLA-4 antibodies. This is a very interesting setup. CTLA-4 blockade therapy is one of the famous checkpoint-based cancer immunotherapies, but it is definitely not the most promising one. Instead, PD-1/L1 blockade is generally considered more promising, so the authors should clarify the reason to exclude PD-1/L1 antibodies in their study.
4. Figure 1C shows that the vaccine given via the s.c. route was more potent than the other approaches, the authors should provide explanations for this interesting finding.
5. One important missing part of the report is the compatibility/safety study of their vaccine formulation. At least histology of the main organs should be done to study the possibility of systemic inflammation reactions.
6. Another question related to this study is that, while it is reported in the literature (e.g. Nature Reviews Materials volume 4, pages415–428(2019)) that nanoparticles with small size (e.g. 10-100 nm) have optimal lymph node drainage, the PLGA particles as the vaccine carrier in the current study are much bigger, so what is the mechanism of particle draining to lymph nodes in the current study?

Point-by-point Response:

Reviewer Comments:

Reviewer #1

Remarks to the Author

The authors present a PLGA-microparticles (MPs)-based cancer vaccine carrying TLR3 agonist Riboxsim and antigens. PLGA-nanoparticles (NPs) or -MPs-based cancer vaccines containing antigens and TLR ligands have been extensively studied by many researchers including the authors. In this manuscript, the authors use Riboxsim, a recently developed TLR3 agonist, and GMP grade PLGA-MPs for possible application to clinical studies. Subcutaneous injection of PLGA-MP-OVA/Riboxsim into mice induced OVA-specific CD8⁺ T cell activation, and retarded tumor growth in EG7-bearing mice. Mouse and human myeloid DCs efficiently take up MP-

OVA/Riboxsim. PLGA-MPs containing Riboxsim and HLA-A0201-restricted cancer peptides induced peptide-specific CD8+ T cell activation in HLA-A0201-transgenic mice. Finally, combination therapy of MP-OVA/Riboxsim and anti-CTLA4 Ab strongly induced tumor regression and prolonged survival in EG7-bearing mice. Overall, these experiments are well-designed and proceeded carefully. However, several important issues remain unsolved.

Major comments

1. The authors mention that Riboxsim is a TLR3 ligand that induces potent DC activation and secretion of type I IFNs (Fig. 3). It is surprising that extracellularly added Riboxsim (100bp dsRNA) strongly induces type I IFN production by BMDCs as compared to polyI:C. Riboxsim is synthesized with DNA/RNA synthesizer or in vitro transcribed? Original paper of Riboxsim (RGC100) by Naumann et al. only shows the data from TLR3 knockdown experiments in JAWSII DCs in terms of TLR3 specificity. To confirm TLR3 specificity of Riboxsim, the authors should examine the cytokine profiles of Riboxsim-stimulated BMDC and CD8alpha+ DCs from TLR3 KO and MAVS KO mice.

Response:

We appreciate your constructive suggestion.

First of all, Riboxsim is produced by incubation of a single-stranded poly cytidyl RNA template with a Calcivirus RNA-dependent RNA polymerase (RdRp) in presence of a mixture of rGTP, rITP, and rCTP.

In response to your suggestion, we performed cytokine profile analysis of Riboxsim-stimulated primary DCs. Similar to previous experiments in C57BL/6 wild-type mice, the secretion of several pro-inflammatory cytokines was also not detectable in Riboxsim-stimulated BMDCs of TLR3^{-/-} and MAVS^{-/-} gene targeted mice. Interestingly, the production of type I IFNs (i.e. IFN α /IFN β) was absent in MAVS-deficient BMDCs, strongly suggesting the activation of the RIG-I/MAVS pathway by Riboxsim via binding to cytosolic nucleic acid receptors. A stimulation of MDA5 by Riboxsim can be ruled out because Riboxsim is with 100 base pairs way too short to stimulate MDA5¹. The exact mechanism of endosomal escape of Riboxsim to engage the cytosolic RNA sensor RIG-I is not known up to date. Also, uptake of dsRNA into the cell via a potential dsRNA-receptor is still a matter of debate. However, polyI:C likely does not leave the endosome. The induction of proinflammatory cytokines by polyI:C as illustrated in Figure 3 is likely attributed to an endotoxin contamination, since endotoxin-free polyI:C did not result in similar pro-inflammatory responses.

Though we did assume TLR3-specific binding of Riboxsim, this notion relied on the original publication of Naumann *et al.*². In fact, the authors used a derivative of Riboxsim designated RGC100. RGC100 encompasses a different base composition in that it doesn't contain IMP in contrast to Riboxsim. As you already mentioned, TLR3 specificity was verified by siRNA-guided knockdown of TLR3 in the DC line JAWS II, while RIG-I/MAVS signaling has not been addressed during the establishment of RGC100 as dsRNA analogue. While being a

preferred TLR3 agonist by the length of the double-stranded RNA, the addition of the 5' triphosphate renders Riboxsim a classical RIG-I ligand as well.

Importantly, stimulation of DCs with Riboxsim and the subsequent activation of RIG-I/MAVS pathway did not induce pro-inflammatory cytokines but resulted in a strong type I IFN response. Several reports have shown that type I IFN production induced by dsRNA is largely attributed to RIG-I/MAVS pathway rather than the TRIF/TICAM-1 pathway³ and that RIG-I mediated type I IFN responses are higher compared to IFN α /IFN β produced after TLR3 signaling by polyI:C⁴. The advantages of type I IFN in induction of cytotoxic T cell responses and antitumor immunity has already been described by numerous studies⁵⁻⁷.

We think, that the combined activation of TLR3/TICAM-1 (for efficient DC maturation and activation to evoke antitumor cellular immunity) and the distinct type I IFN production upon binding to RIG-I is of great advantage for Riboxsim as an ideal adjuvant in cancer immunotherapy.

In the revised manuscript, we have deleted all claims of exclusive TLR3 triggering. Additionally, we added the cytokine profiles of TLR3 and MAVS deficient BMDCs in the supplementary figure section (see new Figure S4a). Furthermore, we also included DC maturation experiments in BMDCs from these KO mice (see new Figure S4b)

Although we are aware that *in vitro* generation of BMDCs using GM-CSF stimulates a higher population of DCs reminiscent of the myeloid lineage phenotype rather than CD8 α + DCs, we could not perform the analysis of the cytokine portfolio of CD8 α + splenic DCs because the generation of CD8 α DCs by magnetic bead separation (#130-091-169, mouse CD8+ Dendritic Cell Isolation Kit, Miltenyi Biotec) did not result in appropriate cell numbers for the suggested experiments in spite of several attempts. As well, we consider maturation and cytokine profile analysis in BMDCs adequate for obtaining comparative results, as BMDCs are used extensively for this purpose in the literature and as we had already performed the previous cytokine secretion experiments with BMDCs (see Figure 3).

2. Several reports have demonstrated that PLGA-NPs are taken up by DCs/macrophages and delivered into both endosomes and cytoplasm where encapsulated molecules are released (Reviewd by Hamdy et al., ADDR, 2011), suggesting that Riboxsim incorporated into PLGA-MPs is transferred to both endosomes and cytosol together with antigens. Thus, it is unknown whether MP-OVA/Riboxsim-mediated CD8+ T cell activation depends on endosomal TLR3. CD8+ T cell activation as well as serum cytokine levels following MP-OVA/Riboxsim injection (s.c.) should be investigated using TLR3 KO and MAVS KO mice.

Response:

We appreciate your comment and experimentally addressed your question. Similar to the cytokine profile in TLR3 and MAVS deficient background, CD8⁺ T cell activation following MP-OVA Riboxsim vaccination was dependent on both pathways. Compared to that, OVA-

specific IFN γ production in response to MP-OVA/polyI:C was reduced in TLR3^{-/-} mice, indicating a prominent dependence on endosomal TLR3.

Importantly, activation of cross-presentation pathways requires endosomal escape, where potential stimulation of cytosolic RIG-I/MAVS signaling is possible. While TLR3 is activated by dsRNA molecules in the endosome after PLGA MP have been internalized by phagocytosis, the exact mechanism by which PLGA particles and encapsulated compounds are escaping into the cytosol still has to be elucidated. However, different theories have been proposed. In general, PLGA particles facilitate endosomal escape of either particles or released cargo. Destabilization of the endosomal membrane may be induced by an alteration of the anionic particle surface charge which enables local interaction with endo-lysosomal membranes. An alternative explanation proposes a “proton-sponge mechanism” resulting in rupture of the endosomal membrane by osmotic pressure during endosomal acidification^{8,9}.

Collectively, co-stimulation of TLR3 and RIG-I by Riboxsim leads to an improved immunostimulatory potency compared to polyI:C for generation of vigorous and antigen-specific CD8⁺ T cell responses without the induction of systemic inflammation. These data have been included as an additional supplementary figure (see new Figure S7). Notably, serum cytokine levels after MP-OVA/Riboxsim vaccination were below the limits of detection.

3. The authors investigated the protective role of MP-OVA/Riboxsim in tumor development using EG7 and MO5 tumor models, but its therapeutic role was examined only in EG7 tumors. In cancer immunotherapy using vaccine adjuvants, therapeutic effect is more important than protective effect. Recent papers by Takeda et al., demonstrate that vaccine immunotherapy with TLR3-specific ligand ARNAX and tumor antigens induces anti-tumor CTLs and enhances anti-tumor responses in conjunction with PD-L1 blockade (Takeda et al., 2017, Cell Repos 19:1874-1887; Takeda et al., Cancer Sci. 2018, 109:2119-2129). Does therapeutic vaccination of MP-OVA/Riboxsim suppress lung metastasis of MO5? These experiments should be done. Also, citation of these articles and discussions are required.

Response:

This recommendation is pertinent. We appreciate our reviewer’s constructive suggestions and agreed to further analyze the therapeutic antitumor activity of MP-OVA/Riboxsim vaccination against aggressive metastatic melanoma. Remarkably, immunization with PLGA MP encapsulating tumor antigen and Riboxsim reduced pulmonary tumor load, which was comparable to the anti-tumor effect induced by MP-OVA/polyI:C. This important data further highlight the efficacy of our system in cancer immunotherapy. Hence, these new data are now shown in Figure S10. As kindly requested, we cited and discussed the two publications by Takeda *et al.* in our revised manuscript.

Minor comments

1. Fig. 5c and Fig. 6b. Tumor growth data of individual mice should be shown in supplemental figures.

Response:

Thank you for this suggestions of displaying individual tumor data. As requested, spider-plot data of tumor growth have been included in the supplementary figures (see Figure S12a,b).

2. Discussion. Among the TLR3 agonists, the authors compare polyICLC and Riboxsim in terms of chemical structure, stability and adverse effects ; polyIC is heterogenous undefined dsRNA (1-8kb) while Riboxsim is 100bp dsRNA. It is wonder that the authors do not mention other TLR3 agonist such as ARNAX that has defined chemical structure and anti-tumor efficacy via activation of professional antigen-presenting DCs (Nat Commun. 6:6280, 2015, Trial Wach, Oncoimmunology, 9:1, 2020). The authors should refer these papers.

Response:

We appreciate the mentioning of suboptimal evaluation of other TLR agonist in the discussion. In response to your helpful criticism, we have carefully revised the manuscript and discussed other immunostimulatory dsRNA molecules that are more comparable to Riboxsim. Especially, the publications about the non-inflammatory TLR3 agonist ARNAX mentioned above have been referred to.

Reviewer #2

Remarks to the Author:

The current manuscript by Groettrup and colleagues reports on the formulation of a cancer vaccine by combining antigens with a TLR3 agonist Riboxsim in PLGA particles. The vaccine was tested in vitro and in vivo, and the results show that its vaccination efficiency in mice was significantly improved compared to that of a control formulation with Poly I:C as the adjuvant. Furthermore, the Riboxsim-based vaccine in combination with CTLA-4 extended the survival of tumor-bearing mice beyond monotherapies with either agent.

Major issues:

1. A major issue of the study is its novelty. Therapeutic or protective cancer vaccines based on adjuvants and antigens in nano/microparticles have been reported in many previous studies, with a high variety of different types of adjuvants and the antigens. The first reports on this topic can be dated back to a few decades ago. Recently there have been a considerable number of publications on combinations of cancer vaccines with other immunotherapeutics such as checkpoint blockade drugs. Besides these previous studies, the team has also published on vaccines containing their own adjuvant Riboxsim. Therefore, the current study is a useful and

informative report on the vaccine developed by the authors, but it brings limited new perspectives to the research field.

Response:

We appreciate your concern. However, this is the first study that characterizes the immunostimulatory and immunomodulatory properties of the dsRNA adjuvant Riboxsim in an anti-tumor vaccine approach. In fact, it is the first publication about Riboxsim altogether. Furthermore, we are the first group to demonstrate the advantage of encapsulated tumor antigens together with Riboxsim and its enhanced therapeutic potential by a combination therapy using specifically anti-CTLA-4 antibody therapy. The original publication of Naumann *et al.* has worked with a derivative of Riboxsim designated RGC100 that contained GMP and CMP only². Also, RGC100 has not been applied in any vaccine formulations or tumor vaccination studies hitherto. In sum, our study provides the first experimental evidence that immune checkpoint blockade with anti-CTLA-4 antibody potentiates PLGA microparticle tumor vaccines by reinvigoration of strong tumor-specific cytotoxic T cell responses and subsequent potent tumor immunity.

2. The design of the current study is non-ideal. One major limitation is that the authors only compared their adjuvant Riboxsim to Poly I:C, which is a known standard TLR3 agonist with limited applicability due to its high toxicity. To really demonstrate the high potential of their adjuvant/vaccine regarding efficacy and safety, the study should involve other TLR agonists or even other types of vaccine adjuvants. There are quite a few vaccine adjuvants on the market, which are not based on TLR agonists. In my mind, those are important controls for this study.

Response:

Thank you for your well taken suggestion of a comparative analysis of other TLR agonists or other types of adjuvants encapsulated into PLGA MPs. In order to demonstrate the high potential of Riboxsim in our system, we analyzed CD8⁺ T cell activation of two other TLR ligands i.e. the TLR4 ligand MPLA (monophosphoryl lipid A) and the TLR7/TLR8 agonist Resiquimod (R848), as well as the CD1 ligand α -Galactosylceramide (α -Gal-Cer, KRN700, Funakoshi, Japan) as a representative adjuvant not based on TLR activation. Compared to the encapsulated Riboxsim, only MP OVA/R848 resulted in similar antigen-specific IFN γ -responses (see new Fig. S9a). Consequently, MP OVA/R848 was used as therapeutic treatment in a subcutaneous EG-7 tumor model. While MP-OVA/Riboxsim induced tumor regression, treatment of tumor-bearing mice with co-encapsulated Resiquimod only led to minor anti-tumor responses similar to MP-OVA/polyI:C. We included these data in the supplementary figures (Figure S9b,c), since it highlights the beneficial use of Riboxsim as a novel dsRNA adjuvant in a PLGA microparticle-based tumor vaccination approach.

3. In the combination therapy study, the vaccine was combined with CTLA-4 antibodies. This is a very interesting setup. CTLA-4 blockade therapy is one of the famous checkpoint-based cancer immunotherapies, but it is definitely not the most promising one. Instead, PD-1/L1 blockade is

generally considered more promising, so the authors should clarify the reason to exclude PD-1/L1 antibodies in their study.

Response:

Thank you; this is a valid concern which we will explain.

We agree that antibody-mediated targeting of PD-1 on T cells or its ligand PD-L1 on tumor tissue or tumor-associated DCs is considered the most potent immune checkpoint inhibitory approach. Blockage of the interaction of PD-1 on the surface of exhausted T cells will transmit signals for T-cell survival and T cell proliferation of already activated T cells. Furthermore, due to the complexity of the tumor microenvironment, the expression of PD-1 and its ligands PD-L1 and PD-L2 can vary in different tumor settings, as well as the response to PD-1 checkpoint inhibition.

By blocking CTLA-4 on the surface of anti-tumor CTLs, CD28 binding to the co-stimulatory molecules CD80/CD86 on the antigen-presenting cells is enabled again and leads to reinvigoration of anti-tumor and cytotoxic T cell effector functions. Attributed to the depot effect after PLGA MP vaccination, dendritic cells constantly foster antigen presentation and T cell priming (“ T cell activation signal 1”) of the encapsulated tumor-associated antigen. Thus, tumor-specific T cells are constantly generated for a prolonged time. Pharmacological blockade of CTLA-4 reinvigorated exhausted antitumor CTLs by providing a pro-survival signal and an additional T cell activation “boost” due to renewed enabling of TCR/CD28 ligation by co-stimulatory molecules on PLGA-MP-triggered mature DCs (“T cell activation signal 2”). Therefore, we postulate a superior and highly synergistic effect of anti-CTLA-4 antibody therapy in combination with the PLGA microparticle vaccine.

In order to experimentally address this issue, we performed a combination therapy of dual anti-CTLA-4/anti-PD-1 antibodies together with PLGA MP-OVA/Riboxim against already established E.G7 tumors. Interestingly, this treatment regimen promoted the antitumor activity of the already potent anti-CTLA-4/PLGA MP therapy. However, the most important part of this synergy has to be attributed to inhibition of CTLA-4. Notably, the addition of PD-1 inhibitors did not provide improvement of the PLGA MP therapy, while anti-PD-1 monotherapy completely failed to induce tumor regression. We have added these data as an additional new figure in the supplementary section (see Figure S11a,b).

4. Figure 1C shows that the vaccine given via the s.c. route was more potent than the other approaches, the authors should provide explanations for this interesting finding.

Response:

Thank you for your positive comment. We will clarify the advantageous use of subcutaneous administration of the PLGA particle vaccine.

First of all, the injectable amount of PLGA particles is the highest when using subcutaneous immunizations, compared to intranodular or intramuscular application. Although,

intramuscular application has advantages, including slow release, low rates of adverse events or injection site reactions, the slow release of encapsulated compounds is ensured by the formation of a depot effect at the subcutaneous injection site, rendering intramuscular application obsolete. Additionally, inferior lymphatic drainage of muscle tissue, as well as low numbers of muscle-resident DCs or other antigen-presenting cells are not ideal for PLGA MP application¹⁰. Similar to muscle tissue, the mouse peritoneal cavity encompasses mainly neutrophils and macrophages. Using s.c. immunizations, we take advantage of the highly populated epidermal and dermal skin layer. Potent antigen-presenting cells such as dermal DCs, Langerhans cell or plasmacytoid DCs possess immunostimulatory and migratory functions for particle uptake, particulate antigen presentation and T cell priming¹¹.

5. One important missing part of the report is the compatibility/safety study of their vaccine formulation. At least histology of the main organs should be done to study the possibility of systemic inflammation reactions.

Response:

We fully agree with your concerns. In order to meet your expectations, the *in vivo* toxicity of our PLGA MP vaccine was assessed by histopathologic analysis of the major organs after immunization with PLGA MP-OVA/Riboxim. No detectable abnormalities were observed by H&E staining in either heart, liver, kidney, lung, spleen, or the intestine. The particle safety profile was included as a new supplementary figure (Figure S5) in the revised manuscript. Consistent with these data, we did not detect serum cytokines after PLGA MP vaccination either (data requested by reviewer 1).

6. Another question related to this study is that, while it is reported in the literature (e.g. *Nature Reviews Materials* volume 4, pages415-428(2019)) that nanoparticles with small size (e.g. 10-100 nm) have optimal lymph node drainage, the PLGA particles as the vaccine carrier in the current study are much bigger, so what is the mechanism of particle draining to lymph nodes in the current study?

Response:

Previously, our group analyzed fluorescent PLGA MP trafficking to draining lymph nodes after dermal footpad injection¹². Analysis of popliteal lymph nodes revealed a substantial amount of “free” PLGA particles that might have entered the lymph nodes by draining lymph flow. Due to the heterogeneity in particle size it might be possible that small PLGA particles are migrating to the draining lymph nodes by lymph flow. Schliehe *et al.* also detected cell-associated PLGA particles in the draining lymph nodes. A phenotypic analysis of the cell types revealed CD8⁻ DCs and macrophages as PLGA MP-positive cells in the lymph node. The accumulation of PLGA MP at the site of injection as a result of the already described depot effect (see Figure 2c) enables local antigen release. Thus, particle drainage

to the lymph node is not obligatory, as antigen is taken up by skin DCs, followed by DC-mediated antigen transport into the draining lymph node.

References cited in the point-to-point response

1. Brisse M, Ly H. Comparative Structure and Function Analysis of the RIG-I-Like Receptors: RIG-I and MDA5. *Front. Immunol.* **10**, 1586 (2019).
2. Naumann, K. *et al.* Activation of dendritic cells by the novel Toll-like receptor 3 agonist RGC100. *Clin. Dev. Immunol.* **2013**, 283649 (2013).
3. Kato, H. *et al.* Differential roles of MDA5 and RIG-I helicases in the recognition of RNA viruses. *Nature* **441**, 101–105 (2006).
4. Linehan, M. M. *et al.* A minimal RNA ligand for potent RIG-I activation in living mice. *Sci. Adv.* (2018) doi:10.1126/sciadv.1701854.
5. Le Bon, A. *et al.* Direct Stimulation of T Cells by Type I IFN Enhances the CD8 + T Cell Response during Cross-Priming . *J. Immunol.* (2006) doi:10.4049/jimmunol.176.8.4682.
6. Thompson, L. J., Kolumam, G. A., Thomas, S. & Murali-Krishna, K. Innate Inflammatory Signals Induced by Various Pathogens Differentially Dictate the IFN-I Dependence of CD8 T Cells for Clonal Expansion and Memory Formation. *J. Immunol.* (2006) doi:10.4049/jimmunol.177.3.1746.
7. Fuertes, M. B. *et al.* Host type I IFN signals are required for antitumor CD8+ T cell responses through CD8 α + dendritic cells. *J. Exp. Med.* **208**, 2005–16 (2011).
8. Panyam, J., Zhou, W.-Z., Prabha, S., Sahoo, S. K. & Labhasetwar, V. Rapid endo-lysosomal escape of poly(DL-lactide-co-glycolide) nanoparticles: implications for drug and gene delivery. *FASEB J.* **16**, 1217–26 (2002).
9. Gros, M. & Amigorena, S. Regulation of antigen export to the cytosol during cross-presentation. *Front. Immunol.* (2019) doi:10.3389/fimmu.2019.00041.
10. Malissen, B., Tamoutounour, S. & Henri, S. The origins and functions of dendritic cells and macrophages in the skin. *Nat. Rev. Immunol.* (2014) doi:10.1038/nri3683.
11. Kashem, S. W., Haniffa, M. & Kaplan, D. H. Antigen-presenting cells in the skin. *Ann. Rev. Immunol.* (2017) doi:10.1146/annurev-immunol-051116-052215.
12. Schliehe, C. *et al.* CD8- Dendritic Cells and Macrophages Cross-Present Poly(D,L-lactate-co-glycolate) Acid Microsphere-Encapsulated Antigen In Vivo. *J. Immunol.* **187**, 2112–2121 (2011).

REVIEWER COMMENTS

Reviewer #1 (Remarks to the Author):

My concerns have been addressed in the revisions.

Reviewer #2 (Remarks to the Author):

The authors have properly addressed the comments of the reviewer with additional data and explanations. It is unfortunate to see that the Riboxim formulation did not enhance the efficacy of PD-1/L1 therapy, however, this should not prevent the publication of the work.

RESPONSE TO REVIEWERS' COMMENTS

Reviewer #1 (Remarks to the Author):

My concerns have been addressed in the revisions.

Response to Reviewer #1: We gratefully thank the reviewer for the final acceptance of all raised questions and concerns. The manuscript has been greatly improved by the reviewer's constructive and thoughtful suggestions and questions.

Reviewer #2 (Remarks to the Author):

The authors have properly addressed the comments of the reviewer with additional data and explanations. It is unfortunate to see that the Riboxxim formulation did not enhance the efficacy of PD-1/L1 therapy, however, this should not prevent the publication of the work.

Response to Reviewer #2:

We appreciate the reviewer for the positive feedback. We were similarly surprised by the outcome of the anti-PD-1 and PLGA MP combination therapy. Hopefully, we have provided reasonable explanation for the importance of CTLA-4 blockage in the PLGA MP mediated anti-tumor response.